A new specimen of Palvennia hoybergeti: implications for cranial and pectoral girdle anatomy in ophthalmosaurid ichthyosaurs

Delsett Lene Liebe 1 l.l.delsett@nhm.uio.no
Druckenmiller Patrick Scott 2 3
Roberts Aubrey Jane 4
Hurum Jørn Harald 1
1 Natural History Museum, University of Oslo , Oslo , Norway
2 University of Alaska Museum , Fairbanks, AK , USA
3 Department of Geosciences, University of Alaska–Fairbanks , Fairbanks, AK , USA
4 Natural History Museum , London , UK
Farke Andrew
Electronic publication date: 2018 Oct 12
Publication date: 2018
Volume: 6
Electronic Location ID: e5776
Received 2018 May 30; Accepted 2018 Sep 15
Copyright: © 2018 Delsett et al.
Copyright year: 2018
Copyright holder: Delsett et al.
License: This is an open access article distributed under the terms of the Creative Commons Attribution License, which permits unrestricted use, distribution, reproduction and adaptation in any medium and for any purpose provided that it is properly attributed. For attribution, the original author(s), title, publication source (PeerJ) and either DOI or URL of the article must be cited.
License URL: https://creativecommons.org/licenses/by/4.0/

Keywords: Ophthalmosauridae, Ichthyosauria, Spitsbergen, Late Jurassic, Pectoral girdle, Skull, Palvennia hoybergeti, PCA

Funding: The Ministry of Education and Research via the Natural History Museum, University of Oslo The Spitsbergen Travel, ExxonMobil, Fugro, Bayerngas Norge, OMV, Helsport, PowerShop and Nexen The Norwegian Research Council, National Geographic EC0425_09 and EC0435_09 The Norwegian Ministry of Education and Research and Brian Snyder Lene Liebe Delsett is supported by a PhD grant from the Ministry of Education and Research via the Natural History Museum, University of Oslo. The sponsors for the excavations were Spitsbergen Travel, ExxonMobil, Fugro, Bayerngas Norge, OMV, Helsport, PowerShop and Nexen. Grants for the excavations were provided by the Norwegian Research Council, National Geographic (grant no. EC0425_09 and EC0435_09), the Norwegian Ministry of Education and Research and Brian Snyder. The funders had no role in study design, data collection and analysis, decision to publish, or preparation of the manuscript.

==============================
The Late Jurassic Slottsmøya Member Lagerstätte on Spitsbergen preserves a diverse array of marine reptiles, including four named taxa of ophthalmosaurid ichthyosaurs. One of these, Palvennia hoybergeti, is based on the single holotype specimen (SVB 1451) with an incomplete skull. A newly discovered specimen (PMO 222.669) with a disarticulated but largely complete skull and anterior postcranium is described, which considerably expands our knowledge of this taxon. Two additional new ophthalmosaurid specimens with pectoral girdles from the same member are described. The taxonomic utility of the ophthalmosaurid pectoral girdle is contentious, and an assessment of seven pectoral girdles from the Slottsmøya Member provides a basis for addressing this question via a 2D landmark principal component analysis of baracromian coracoids. The analysis reveals a taxonomic signal in the coracoids but also highlights the degree of individual variation. Commonly used phylogenetic characters do not fully encapsulate the degree of variation seen in coracoids and in some cases combine analogous features.

Introduction

Ichthyopterygia was one of the major secondarily aquatic vertebrate clades that existed during the Mesozoic. Fossils are found from the Early Triassic (Olenekian) to the early Late Cretaceous (Cenomanian). The Late Jurassic was an interval with great species richness of thunnosaurian ichthyosaurs that evolved an elongated skull with an enormous eye and forefins larger than hindfins (Bardet, 1992; Motani, 2005; Fischer et al., 2016).

The Slottsmøya Member Lagerstätte (SML) on Spitsbergen, Norway (Figs. 1 and 2), is well known for its abundance of marine reptile remains (ichthyosaurs and plesiosaurians) from the latest Jurassic and the earliest Cretaceous (Hurum et al., 2012; Delsett et al., 2016). Four new monospecific ophthalmosaurid genera and three undetermined species of ophthalmosaurids have recently been described (Druckenmiller et al., 2012; Roberts et al., 2014; Delsett et al., 2017). Another 19 additional specimens have been collected, and three of those specimens are described in this paper. The most complete, PMO 222.669, consists of an anterior portion of an individual that we refer to the SML species Palvennia hoybergeti (Druckenmiller et al., 2012) based on its similarity to the holotype specimen (SVB 1451) in their overlapping material. The new specimen greatly increases our knowledge regarding the anatomy of this species, including new information from the skull as well as the first information regarding the morphology of the pectoral girdle and forefins. Additionally, two disarticulated and incomplete ophthalmosaurid specimens, PMO 222.658/PMO 230.097 (material from the same specimen, see Materials and Methods) and PMO 224.250, are described.

Figure 1 Map showing the discovery sites of the three ichthyosaur specimens described herein and the holotype specimen (SVB 1451) of P. hoybergeti (red dots).

The reference map (white) shows the Svalbard archipelago and the excavation area (orange dot) on the main island Spitsbergen. Adapted from Hurum et al. (2012).

Figure 2 Composite section of the Slottsmøya Member Lagerstätte showing the vertical distribution of ophthalmosaurid specimens with preserved pectoral girdles discussed in the text.

New specimens described in this contribution in bold. Modified from Delsett et al. (2017) and Koevoets (2017).

Combined with currently described material from the locality, the new specimens offer insight into pectoral girdle variation of ophthalmosaurid ichthyosaurs. The homology of certain features on the elements of the ichthyosaurian pectoral girdle as well as its architecture in life was the subject of heated early debates (Home, 1818; Seeley, 1874, 1893; Hulke, 1892), but is now known to be relatively similar to terrestrial reptiles (Johnson, 1979; Sander, 2000), consisting of a pair of coracoids and scapulae, a pair of clavicles and an interclavicle. As is common in aquatic tetrapods, the scapula is not fused to the coracoid (Sander, 2000). Historically, the iconic species Ophthalmosaurus icenicus was erected on the basis of characters in the pectoral girdle and forefin (Seeley, 1874; Moon & Kirton, 2016), but the actual taxonomic utility of pectoral girdle elements in post-Triassic ichthyosaurs has been questioned (McGowan, 1974; Johnson, 1979; Druckenmiller & Maxwell, 2010; Moon & Kirton, 2016). A challenge is the subjective assessment of individual variation, as many taxa are described from a single specimen. In this study, we use geometric morphometrics on coracoids to investigate the range of individual variation and phylogenetic signal.

Geological Setting

The Slottsmøya Member is one of four members of the Agardhfjellet Formation, in the Janusfjellet Subgroup of the Adventdalen Group (Fig. 2). The SML consists of Tithonian to Berriasian-aged sediments that crop out north of the town of Longyearbyen on Spitsbergen, the largest island in the Svalbard archipelago, located between 74° and 81° North and 10° and 35° East. Deposition of the unit occurred broadly in an offshore transition when Svalbard was located farther south, at a paleolatitude of 63°–66° North (Torsvik et al., 2012). The sedimentology and stratigraphy of the Agardhfjellet Formation is described in detail in other contributions (Collignon & Hammer, 2012; Dypvik & Zakharov, 2012; Hammer, Collignon & Nakrem, 2012; Koevoets et al., 2016, 2018; Koevoets, 2017), but the Slottsmøya Member is a 70–100 m thick, upward-coarsening unit made up primarily of dark-gray to black shales, paper shales and siltstones with higher invertebrate abundance than in the other members in the formation (Collignon & Hammer, 2012; Koevoets, 2017). The zero m level in the section is set at a remarkable echinoderm-rich bed laterally continuous throughout the study area and probably representing a storm deposit (Rousseau & Nakrem, 2012; Rousseau, Gale & Thuy, 2018).

The oxygenation fluctuated repeatedly during the deposition of the member, with periodic oxygenation of the bottom waters (Collignon & Hammer, 2012; Koevoets, 2017). Two of the three specimens described here, PMO 222.669 and PMO 224.250, were found in the part of the section with the highest abundance of vertebrate remains (10–20 m above the echinoderm bed), many of which are articulated or partly articulated (Delsett et al., 2016) (Fig. 2). PMO 222.669 was partly covered in bivalves and crinoids situated on the actual skeletal elements, similar to a previously described specimen (PMO 222.670; Ophthalmosauridae indet.) (Delsett et al., 2017), and as the specimens were found only one m apart stratigraphically, this could represent an event with more oxic bottom conditions. However, significant bivalve abundances are also found in parts of the section with relatively low oxygen levels (Koevoets, 2017), and total organic content seems to covary more closely with the degree of articulation of the vertebrates than does oxygenation (contra Delsett et al., 2016).

In the uppermost part of the member, a sudden change in the environment is recorded through significant changes in the invertebrate and teleost fauna (Koevoets, 2017). In this part of the section, just above one of the specimens described herein (PMO 222.658), 15 methane seeps were described (Hryniewicz et al., 2015). Overall, the preservation of the vertebrates in the entire member is more three dimensional than would normally be expected with the known compaction rate (Hammer, Collignon & Nakrem, 2012), likely as a consequence of early barite permineralization in many of the elements, which in turn is probably a result of dissolution of barite in the sea water due to methane seepage (Delsett et al., 2016).

Materials and Methods

PMO 222.658/PMO 230.097 is an ichthyosaur specimen from which elements were collected two different years: in 2009 and 2010, and the elements were originally catalogued separately. From here on out, the museum number given first (PMO 222.658) will be used in the text. The ichthyosaur specimen PMO 222.669 was collected in 2011 and PMO 224.250 in 2012. The stratigraphic position of the specimens was determined with a total station. The specimens were collected in protective plaster jackets and transported to NHM in Oslo for mechanical preparation. PMO 222.669 was largely covered in siderite, gypsum, and barite, and an air scribe and a sand blaster were used to remove the matrix. The taphonomy and stratigraphic position of the specimens has previously been discussed in Delsett et al. (2016). An additional unassigned ophthalmosaurid specimen that is under description, PMO 222.667, is included for comparative purposes, as it was in Roberts et al. (2014). The specimens are housed in the palaeontological collections of Natural History Museum, Oslo. The most commonly used chronostratigraphic names (e.g., Tithonian) will be used in this work instead of the regional names (e.g., Volgian). The following permits were given by the Governor of Svalbard for the excavations in 2009, 2010, 2011, and 2012: 2006/00528-13, RIS ID 3707; RIS ID: 4760 and 2006/00528-39.

The taxonomic utility of the post-Triassic ichthyosaurian pectoral girdle has been challenged (Maxwell & Druckenmiller, 2011; Lomax, 2017). Because seven SML specimens with relatively well-preserved pectoral girdles have now been recovered, it provides an opportunity to test the assertion that there is little taxonomic signal in the pectoral girdles. Here, we conduct a principal component analysis (PCA) on 2D landmarks of coracoids from a larger set of baracromian ichthyosaurs including six of the SML specimens. Geometric morphometrics is well suited for evaluating shape difference, and coracoids are frequently preserved in their entirety (Lomax, 2017) with a well-defined outline in dorsal and ventral view making them suited for a PCA on 2D landmarks. In contrast, ichthyosaurian scapulae have processes arranged in orthogonal planes, which reduces the efficiency of 2D methods to capture the fine variations of their morphology.

To understand the degree of individual variation, one aim of the analysis is to determine whether Ophthalmosaurus and Stenopterygius coracoids can be separated on the basis of this method. The two genera were chosen for this purpose as they are known from multiple specimens. Coracoid landmarks were digitized from photographs either taken by the authors in museum collections or provided by colleagues. Only adult specimens where all landmarks could be scored were included. Photographs were taken from directly above the element to reduce distortion. Both left and right coracoids were used and mirrored to ensure comparable values. Six landmarks were selected (Fig. 3) with an aim of maximizing the quantity of shape information based on points used in descriptions and/or phylogenetic characters. Landmarks 2–6 are homologous points based on contact between the coracoid and other skeletal elements, that is, anterior and posterior ends of facets for the other coracoid, the scapula and the humerus. Landmark 1 represents the anterolateral corner of the anteromedial process of the coracoid, an often used feature in descriptions and in phylogenetic analysis. Coordinates for each landmark were found using Adobe Photoshop (CS6 ver. 13). The analysis was run in PAST (Hammer, Harper & Ryan, 2001) and included a Procrustes fitting of the values before the PCA. To investigate the properties of each principal component, the Deformations function was used, which shows how the landmarks act as the PC values change. In addition, a discriminant analysis was run between two assigned groups: Ophthalmosaurus and Stenopterygius. A MANOVA test was run to test the groups, set to four constraints.

Figure 3 Landmarks used in the PCA.

The landmarks are shown on an outline of a representative coracoid (PMO 222.669) as an example. Numbered points: (1) Anterolateral corner of the anteromedial process. (2) Anterior end of the intercoracoid facet. (3) Posterior end of the intercoracoid facet. (4) Posterior end of the glenoid contribution. (5) Border between glenoid contribution and scapular facet. (6) Anterior end of scapular facet.

A total of 30 specimens were included in the analysis (Table 1), with the majority represented by two genera, Ophthalmosaurus and Stenopterygius. Additional specimens include ophthalmosaurid species for which only a single specimen is known and three undetermined ophthalmosaurids from the SML. Stenopterygius material was not separated on a sub-generic level in the analysis as no pectoral girdle characters are currently used to differentiate at a species level (Johnson, 1979; Maxwell, 2012). In addition, the inclusion of the non-ophthalmosaurid genus Stenopterygius was to investigate whether it can be clearly separated from ophthalmosaurids. Certain taxa with known preserved coracoids (Platypterygius australis, Nannopterygius enthekiodon, and Baptanodon natans) were not included in this study due to a lack of data or insufficient photo quality.

Table 1 Baracromian specimens used in the PCA on 2D landmarks on coracoids.

Abbrev.	Museum number	Taxon	Photo	
Ac	CMN 40608	Arthropterygius chrisorum	E. Maxwell	
Ad	SNHM 1284-R	Acamptonectes densus	V. Fischer	
H	SMSS SGS	Platyperygius hercynicus	R. Vanis	
Jl	PMO 222.654	Janusaurus lundi	AJR	
Oi1	CAMSM J65583	Ophthalmosaurus icenicus	LLD	
Oi2	NHMUK R3013	Ophthalmosaurus icenicus	AJR	
Oi3	LEICT 100 1949 20	Ophthalmosaurus icenicus	LLD	
Oi4	NHMUK R2137	Ophthalmosaurus icenicus	AJR	
Oi5	OUMNH J48008	Ophthalmosaurus icenicus	LLD	
Oi6	CAMSM J66275	Ophthalmosaurus icenicus	LLD	
Oi7	LEICT 100 1949 18	Ophthalmosaurus icenicus	LLD	
Oi8	CAMSM J29809	Ophthalmosaurus icenicus	LLD	
Oi9	CAMSM J65813	Ophthalmosaurus icenicus	LLD	
Oi10	LEICT 100 1949 2	Ophthalmosaurus icenicus	LLD	
Oi11	CAMSM J29807	Ophthalmosaurus icenicus	LLD	
Oi12	LEIUG 90986/913	Ophthalmosaurus icenicus	LLD	
P	UPM EP-II-8 (1076)	Paraophthalmosaurus kabanovi	M. Arkhangelsky	
Ph	PMO 222.669	Palvennia hoybergeti	LLD	
S1	SMNS 81961	Stenopterygius quadriscissus	LLD	
S2	SMNS 55074	Stenopterygius sp.	LLD	
S3	SMNS 51142	Stenopterygius quadriscissus	LLD	
S4	BSPGHM S.q.	Stenopterygius quadriscissus	LLD	
S5	SMNS 57532	Stenopterygius uniter	LLD	
S6	SMNS 50165	Stenopterygius quadriscissus	LLD	
Si	IRSNB R269	Sveltonectes insolitus	V. Fischer	
SML1	PMO 224.250	Ophthalmosauridae indet., SML	LLD	
SML2	PMO 222.667	Ophthalmosauridae indet., SML	LLD	
SML3	PMO 222.658	Ophthalmosauridae indet., SML	LLD	
U1	UPM EP-II-20 (572)	Undorosaurus gorodischensis	N. Zverkov	
U2	PMO 214.578	Cryopterygius kristiansenae	LLD	

Systematic Palaeontology

Ichthyosauria De Blainville, 1835

Neoichthyosauria Sander, 2000

Thunnosauria Motani, 1999

Ophthalmosauridae Baur, 1887

Palvennia hoybergeti Druckenmiller et al., 2012

New material of the holotype specimen (Fig. 4)

Remark: Several elements from the P. hoybergeti holotype (SVB 1451) that were not included in the original description (Druckenmiller et al., 2012) were discovered during a more careful examination of the material. These elements include the articulars, the second jugal, an assemblage of forefin elements and some very poorly preserved and partial elements from the pectoral girdle and forefins. The articulars and zeugo- and autopodial elements will be described here as they represent overlapping material with the new specimen described below.

Figure 4 Articulars and forefin elements of SVB 1451, holotype of P. hoybergeti.

Left articular in (A), medial view and right articular in (B), medial view. Zeugo- and autopodial elements in (C), dorsal (or ventral) view. Radius and intermedium in articulation, the other elements are disarticulated. Abbreviations: art, articular end; dc, distal carpal; im, incomplete margin; in, intermedium; mc, metacarpal; p, phalanx; pi, pisiform; R, radius. Scale bar = 50 mm. Photo: Lene Liebe Delsett.

Articular Figs. 4A and 4B

Both articulars are preserved (Figs. 4A and 4B). They are mediolaterally compressed taphonomically, and the left articular is incomplete. The articular is clearly anteroposteriorly longer than dorsoventrally tall, as in Janusaurus lundi (Roberts et al., 2014). In O. icenicus the anteroposterior length is only slightly longer or equal in length (LEIUG 90986, MANCH L10301, L. L. Delsett & A. J. Roberts, 2015, personal observation), whereas in Mollesaurus periallus it is dorsoventrally taller than anteroposteriorly long (Fernández, 1999). In anterior view, the articular surface is directed anteriorly and medially with a triangular outline, as in Acamptonectes densus (Fischer et al., 2012). The lateral margin of this surface is concave, the dorsomedial margin is convex and the ventromedial margin is concave. In contrast, the medial side of the element is convex. A thin flange extends ventrally from the anteroposterior midpoint. In medial view, the dorsal margin of the element is slightly concave, in contrast to Sisteronia seeleyi (Fischer et al., 2014b), Arthropterygius chrisorum and Platypterygius australis (Kear, 2005; Maxwell, 2010), all of which are narrowest posteriorly and dorsally convex. The posterior end is not mediolaterally thickened in comparison to the middle of the element, in contrast to Platypterygius australis (Kear, 2005). The lateral surface, which articulates with the surangular, is flat.

Zeugo- and autopodial elements (Fig. 4C)

Nine disarticulated zeugo- and autopodial elements are present in the holotype (Fig. 4C). They are interpreted based on their similarities with PMO 222.669 and are strikingly similar to the latter. The elements are interpreted as a radius, an intermedium, a pisiform, distal carpal 3, three metacarpals, and two phalanges. The element interpreted to be the pisiform is relatively larger than in PMO 222.669, otherwise the relative sizes and morphologies are similar.

***

Referred material: PMO 222.669, a partially articulated and almost complete anterior half of a moderately large ichthyosaur (Figs. 5–13; Table 2; Table S1).

Figure 5 Skeletal map of PMO 222.669, newly referred specimen of P. hoybergeti, in ventral view (stratigraphically up).

Only elements visible in this view are included. Abbreviations: aa, atlas-axis; an, angular; ann, anterior notch; ar, articular; bo, basioccipital; c, coracoid; cl, clavicle; dt, dentary; fr, frontal; h, humerus; hy, hyoid; j, jugal; icl, interclavicle; n, nasal; op, opisthotic; pmx, premaxilla; pt, pterygoid; pof, postfrontal; pra, prearticular; q, quadrate; s, upratemporal; sa, surangular; sca, scapula; sp, splenial; st, stapes; v, vomer. Black triangles = teeth. Scale bar = 50 cm. Modified and corrected from Delsett et al. (2016).

Figure 6 Rostrum and teeth of PMO 222.669, referred specimen of P. hoybergeti.

(A), photograph and (B), interpretation of the rostrum from the surface stratigraphically down. Disarticulated teeth in (C), and (D), different views of the same tooth and (E), and (F), different views of a second tooth. Abbreviations: dt, dentary; fpmx, fossa premaxillaris; l, lacrimal; n, nasal; pmx, premaxilla; sc, sclerotic plate; sp, splenial. Gray triangles = teeth. Scale bar for (A, B) = 100 mm and (C–F) = 10 mm. Photo: Lene Liebe Delsett.

Figure 7 Skull roof of PMO 222.669, referred specimen of P. hoybergeti.

(A), photograph and (B), interpretation of the skull roof in dorsal view. Abbreviations: bo, basioccipital; ex, exoccipital; fm, foramen magnum; fr, frontal; n, nasal; op, opisthotic; pa, parietal; paf, parietal foramen; pf, prefrontal; pof, postfrontal; s, supratemporal; so, supraocciptal; suf, supratemporal fenestra; t, tubercles; ?, uncertain suture. Scale bar = 100 mm. Photo: Lene Liebe Delsett.

Figure 8 Cranial elements of PMO 222.669, referred specimen of P. hoybergeti.

(A), right postorbital in lateral view. (B), left jugal in medial view. (C), left lacrimal in lateral view. (D), left vomer in lateral view. (E), left pterygoid in ventral view. (F), hyoid in unknown orientation. Abbreviations: ap, anterior process; d, depression; dp, dorsal process; im, incomplete margin; hr, horizontal ridge; hs, horizontal shelf; lm, lateral process; lp, lateral process; ma, matrix; mp, medial process; pab, posteriorly ascending bar; pf, prefrontal; qr, quadrate ramus; sb, suborbital bar; vr, vertical ridge. Scale bar = 50 mm. Photo: Lene Liebe Delsett.

Figure 9 Basicranium elements of PMO 222.669, referred specimen of Palvennia hoybergeti.

Basioccipital in (A), posterior, and (B), dorsal view with articulated exoccipitals and (C), lateral and (D), ventral view. Basisphenoid in (E), ventral view. Anterior to the top. Right opisthotic in (F), posterior view. Right stapes in (G), posterior view. Left quadrate in (H), posterior and (I), ventral view. Abbreviations: ac, articular condyle; af, articular facet; bpp, basipterygoid process; eca, extracondylar area; ex, exoccipital; ffm, floor of foramen magnum; icf, intercarotid foramen; im, incomplete margin; mp, medial process; np, notochordal pit; of, opisthotic facet; ol, occipital lamella; pl, pterygoid lamella; pop, paroccipital process; ps, parasphenoid; qjf, quadratojugal facet; saf, surangular facet; stf, stapedial facet. Scale bar = 50 mm. Photo: Lene Liebe Delsett.

Figure 10 Mandibles of PMO 222.669, referred specimen of P. hoybergeti.

Right articular in (A), medial, (B), lateral and (C), articular view. Right mandible with quadrate in (D), lateral view. Left mandible in (E), medial view and (F), interpretation. Left angular in (G), medial and (H), lateral view. Abbreviations: ar, articular; fsa, fossa surangularis; gf, glenoid fossa; im, incomplete margin; lf, lateral flange; MAME, M. adductor mandibulae extremus process; Mc, symphyseal portion of Meckelian canal; mf, medial flange; pap, paracoronoid process; pra, prearticular; q, quadrate; sa, surangular; sc, sclerotic plate; sf, surangular foramen. Scale bar for (A–C) = 25 mm and for (D–H) = 100 mm. Photo: Lene Liebe Delsett.

Figure 11 Pectoral girdle of PMO 222.669, referred specimen of P. hoybergeti.

Interclavicle and clavicles in (A), ventral view. Articular surface of left scapula in (B), anterior view. Right scapula in (C), lateral and (E), medial view. Left scapula in (D), lateral and (F), medial view. Coracoids in (G), ventral and (H), dorsal view. Abbreviations: acp, acromion process; amp, anteromedial process; ann, anterior notch; cf, coracoid facet; cl, clavicle; gf, glenoid facet; incf, intercoracoid facet; ms, median stem; scaf, scapular facet; tb, transverse bar. Scale bar for (A), (F–G) = 100 mm and for (B–E) = 50 mm. Photo: Lene Liebe Delsett.

Figure 12 Forefins of PMO 222.669, referred specimen of P. hoybergeti.

Right forefin in (A), dorsal view (B), interpretation and (C) ventral view. Left humerus in (D), ventral, (E), anterior and (F), posterior view. Abbreviations: dc, distal carpal; dp, dorsal process; dpc, deltopectoral crest; in, intermedium; mc, metacarpal; p, phalanx; pae, preaxial accessory element; paef, facet for preaxial accesory element, path, pathological feature; R, radius; rd, radiale; t, tubercles; U, ulna; Uf, facet for ulna; ul, ulnare. Scale bar = 50 mm. Photo: Lene Liebe Delsett.

Figure 13 Vertebral column of PMO 222.669, referred specimen of P. hoybergeti.

Atlas-axis in (A), anterior, (B), left lateral and (C), ventral view (anterior to the left). Articulated anterior dorsal vertebrae in (D), dorsal view (anterior to the left). Articulated dorsal vertebrae in (E), lateral view (anterior to the right). Abbreviations: dat, diapophysis on atlas; dax, diapophysis on axis; pat, parapophysis on atlas; pax, parapophysis on axis; vca, ventral concave area. Scale bar for (A–C) = 25 mm and for (D, E) = 100 mm. Photo: Lene Liebe Delsett.

Table 2 Selected measurements in millimeters from the forefin and pectoral girdle in the three described ophthalmosaurid specimens.

	PMO 222.669	PMO 222.658	PMO 224.250	
Humerus proximodistal length	167	104	195	
Humerus anteroposterior length proximal end	109	70	142	
Humerus anteroposterior length distal end	120	63	149	
Radius proximodistal length	46	20	52	
Radius anteroposterior width	59	25	68	
Ulna proximodistal length	53	30	57	
Ulna anteroposterior width	61	36	57	
Preaxial accessory element proximodistal length	45	NA	49	
Preaxial accessory element anteroposterior width	41	NA	51	
Scapula proximodistal length	215	NA	NA	
Scapula dorsoventral height proximal blade	NA	NA	175	
Coracoid mediolateral width	220	110	240	
Coracoid anteroposterior length	260	150	270	
Clavicle proximodistal length	NA	NA	NA	
Note:

If both the right and the left element are preserved, the largest element is listed. Additional measurements in Tables S1–S3.

Locality: Island of Spitsbergen, north side of Janusfjellet, approximately 13 km north of Longyearbyen, Svalbard, Norway. UTM WGS84 33X 0519622 8695649.

Horizon and stage: Slottsmøya Member, Agardhfjellet Formation, Janusfjellet Subgroup, late Tithonian, Late Jurassic. 15.5 m above the echinoderm marker bed (Delsett et al., 2016).

Emended differential diagnosis for P. hoybergeti

Moderately large ophthalmosaurid ichthyosaur with one autapomorphy (shown with *) and unique character combination: relatively short rostrum with snout ratio of 0.59 (relatively longer in Caypullisaurus bonapartei and more gracile in Aegirosaurus leptospondylus, N. enthekiodon (Kirton, 1983; Bardet & Fernández, 2000; Fernández, 2007)); very large orbit (comparatively smaller in Cryopterygius kristiansenae, Brachypterygius extremus, Caypullisaurus bonapartei (McGowan, 1976; Fernández, 1997; Druckenmiller et al., 2012)); strongly bowed jugal (relatively straight in Cryopterygius kristiansenae, Brachypterygius extremus (McGowan, 1976; Druckenmiller et al., 2012)); narrow postorbital bar (broad in Cryopterygius kristiansenae, Caypullisaurus bonapartei (Fernández, 2007; Druckenmiller et al., 2012)); frontals mediolaterally broad on skull roof (little frontal exposure on skull roof in Athabascasaurus bitumineus, O. icenicus (Druckenmiller & Maxwell, 2010; Moon & Kirton, 2016)); long frontal-postfrontal contact (short in Platypterygius australis, O. icenicus (Kear, 2005; Moon & Kirton, 2016)); *very large pineal foramen; posterolateral process on pterygoid present (absent in Platypterygius australis (Kear, 2005)); extracondylar area of basioccipital not visible in posterior view (visible in O. icenicus, Sveltonectes insolitus, Acamptonectes densus (Fischer et al., 2011, 2012; Moon & Kirton, 2016)) and lacking a ventral notch (ventral notch in O. icenicus, Mollesaurus periallus (Fernández, 1999; Moon & Kirton, 2016)); lateral and ventral surface of basioccipital with broad extracondylar area of finished bone (very narrow extracondylar area in Sveltonectes insolitus, Brachypterygius extremus (McGowan, 1976; Fischer et al., 2011)); anterior face of basioccipital lacks notochordal pit and basioccipital peg (present in Arthropterygius chrisorum (Maxwell, 2010)); short and robust paroccipital process of opisthotic (elongated in O. icenicus, Acamptonectes densus (Fischer et al., 2012; Moon & Kirton, 2016)); gracile and constricted stapedial shaft (robust shaft without constriction in Platypterygius australis and O. icenicus (Kear, 2005; Moon & Kirton, 2016)); anteromedial process of coracoid present (absent in Platypterygius australis, P. hercynicus, and Caypullisaurus bonapartei (Kear, 2005; Fernández, 2007; Kolb & Sander, 2009)); medial stem of the interclavicle is longer than the transverse bar (transverse bar longer in Caypullisaurus bonapartei and Aegirosaurus leptospondylus (Bardet & Fernández, 2000; Fernández, 2007)); facet for preaxial accessory element (absent in Cryopterygius kristiansenae, N. enthekiodon, Sveltonectes insolitus (Kirton, 1983; Fischer et al., 2011; Druckenmiller et al., 2012)); proximal end of humerus slightly wider than distal end in dorsal view (distal end wider in O. icenicus, Cryopterygius kristiansenae, Platypterygius australis (Kear, 2005; Druckenmiller et al., 2012; Moon & Kirton, 2016)); absence of humerus-intermedium contact (contact in Brachypterygius extremus, Maiaspondylus lindoei, and Aegirosaurus leptospondylus (Kirton, 1983; Bardet & Fernández, 2000; Maxwell & Caldwell, 2006)); phalanges rounded and not tightly packed (rectangular and tightly packed in Cryopterygius kristiansenae, Sveltonectes insolitus, Platypterygius australis (Kear, 2005; Fischer et al., 2011; Druckenmiller et al., 2012)).

Preservation

Based on the orientation of the articulated skull roof, braincase, and vertebrae as preserved in the excavation site, PMO 222.669 landed on its dorsal side on the sea floor after death (Fig. 5; Delsett et al., 2016). The specimen was covered with crinoid, ophiuroid, and bivalve remains, possibly indicating prolonged exposure on the sea bottom in oxygenated bottom waters as an “ichthyosaur fall,” and scavenging might be a cause for the partly disarticulated status (Dick, 2015; Delsett et al., 2016). The elements of the rostrum (splenials, dentaries, and premaxillae) and the anterior part of the nasals are preserved together with the teeth anteriorly. Posterior to the rostral elements is a small disruption of the layers due to permafrost with elements from the palate and skull roof. The nasals and frontals are relatively well preserved, but some other remains in this area are not discernable and probably include the prefrontals and maxillae. The skull roof is preserved in articulation with the basicranium. The left surangular is preserved in articulation with the prearticular and the articular on the right side of the skull, and is turned 180° posteriorly. In close proximity are both angulars. The right surangular is complete but distorted. The atlas-axis complex is preserved along with 24 presacral vertebral centra found in articulation with dorsal ribs in two short series. The centra are three dimensionally preserved and not compressed. Small pieces from as many as three vertebral centra are preserved directly posterior to the axis. Preservational factors preclude most measurements. The specimen preserves only a few neural arches, including that of the atlas axis, as well as some broken gastralia. Two humeri are preserved in articulation with the zeugopodium and some autapodial elements. The left forefin is dorsoventrally compressed and the description is based primarily on the right forefin. Although the scapulae are reasonably three dimensional and nearly complete, the left scapula has suffered some deformation. The clavicle-interclavicle complex is deformed and fractured.

Ontogeny

PMO 222.669 is interpreted to be an adult specimen. The wings on the pterygoid quadrate ramus are robust and of relatively similar size, and the basipterygoid processes are distinct, both indications of an adult stage (Fischer et al., 2011; Kear & Zammit, 2014). This is supported by well-developed distal facets on the humerus and a finished bone surface on all skeletal elements (Johnson, 1977). The specimen is of the same size as the adult holotype of P. hoybergeti and is relatively large compared to other SML specimens, which also support an adult stage (Druckenmiller et al., 2012; Roberts et al., 2014; Delsett et al., 2017).

Premaxilla (Fig. 6)

The preservation of the premaxillae in PMO 222.669 is superior to that seen in the holotype of P. hoybergeti (SVB 1451) (Druckenmiller et al., 2012). They are preserved with their medial contact intact dorsally, but the ventral margins are rotated laterally due to dorsoventral compression. The element is rounded in cross-section anteriorly and more mediolaterally compressed posteriorly, where it also increases in dorsoventral height. The dorsal margin is mediolaterally very narrow in anterior view. In lateral view, two large anteroposteriorly elongated foramina in the anterior portion develop into a groove posteriorly (fossa premaxillaris), approximately at the same point as in O. icenicus and Cryopterygius kristiansenae (Druckenmiller et al., 2012; Moon & Kirton, 2016). The lateral and medial walls of the dental groove are of the same dorsoventral height except at its posteriormost end, where the lateral wall extends further ventrally as in Platypterygius hercynicus (Fischer, 2012). In the anterior portion of PMO 222.669 some slight depressions on the medial side of the dental groove might represent distinct alveoli (Fischer et al., 2012).

Nasals, frontals, parietals, and postfrontals (Fig. 7)

The skull roof preserves most interelement relationships, although some sutures are difficult to discern. The nasals border the frontals anteriorly and laterally. As preserved, the left nasal is covered laterally by the pterygoid. The elongated anterior portions of both nasals continue through the fault zone, but because of distortion, the true anterior length is unknown. The lateral portion of the nasal curves ventrally at a 90° angle compared to the dorsal surface. The skull roof is conspicuously similar to the holotype (SVB 1451) in having an unusually large parietal foramen bordered by the frontals anteriorly and laterally and the parietals posteriorly, which is autapomorphic for this taxon. The frontals do not contact the supratemporal fenestra, which is similar to the situation in Athabascasaurus bitumineus, O. icenicus, and Leninia stellans (Druckenmiller & Maxwell, 2010; Fischer et al., 2013; Moon & Kirton, 2016) but different from many platypterygiine ophthalmosaurids (Fischer et al., 2011, 2012).

The postfrontals border the supratemporal fenestra laterally and contact the supratemporals posteriorly and the frontals anteriorly. The dorsoventrally thickest portion of the postfrontal is in the region between the anterior and posterior portions, as in Platypterygius australis (Kear, 2005). Both postfrontals are preserved in articulation. The left is fairly complete and could be taken out for study. Anteriorly it is articulated to the posterior part of the prefrontal, which is oval in cross-section. The posterior portion is mediolaterally wider than in Platypterygius australis (Kear, 2005). In dorsal view, the shape is similar to that of the holotype (Druckenmiller et al., 2012). The anterior margin is covered by the narrow fingerlike projection of the posterior margin of the nasal. The lateral flange is very thin as in O. icenicus (Moon & Kirton, 2016). The medial margin of the element forms the lateral wall of the supratemporal fenestra and is covered by the projecting finger of the supratemporal in somewhat more than the posterior half of the anteroposterior length of the foramen. The supratemporal finger covers less area than in L. stellans (Fischer et al., 2013). The medial margin is largely concave as in O. icenicus (Moon & Kirton, 2016), but the ridge demarcating the ventral border of the concavity is diagonal and extends much further laterally in this specimen, making a larger concavity.

Postorbital (Fig. 8A)

In PMO 222.669 both postorbitals are preserved. The right postorbital is nearly complete, but is missing the dorsal-most and ventral-most portions of the element. Based on the holotype specimen (SVB 1451), the postorbital bar in P. hoybergeti is known to be dorsoventrally tall and mediolaterally narrow (Druckenmiller et al., 2012). The postorbital is anteroposteriorly narrow compared to Cryopterygius kristiansenae (Druckenmiller et al., 2012). The anterior margin is more strongly curved than Platypterygius australis (Kear, 2005) and instead resembles L. stellans (Fischer et al., 2013). The cross-section of the dorsal-most two-thirds of the element is triangular. It has a pronounced dorsoventrally oriented ridge near the dorsoventral midpoint that is anteriorly inclined as in L. stellans (Fischer et al., 2013). From the ridge, the element tapers toward the anterior and posterior margins. The posterior margin does not have the pronounced posteroventral heel found in O. icenicus and Platypterygius australis (Kear, 2005; Moon & Kirton, 2016). The element is mediolaterally flatter in the more ventral portion as in O. icenicus (Moon & Kirton, 2016). In lateral view, the surface texture of the anterior portion is dorsoventrally striated while the posterior portion is less smooth and instead lightly wrinkled, the latter possibly representing an area for contact with the quadratojugal. Medially, only the dorsalmost preserved part is striated, possibly for articulation with the squamosal or the supratemporal.

Jugal (Fig. 8B)

A complete left and an incomplete right jugal are preserved, which strongly resemble that of the holotype specimen of P. hoybergeti (SVB 1451) (Druckenmiller et al., 2012). In overall morphology the jugal is strongly dorsally curved as in O. icenicus and Paraophthalmosaurus (“Yasykovia”), but in contrast to Cryopterygius kristiansenae, which is nearly straight (Arkhangelsky, 1997; Efimov, 1999b; Druckenmiller et al., 2012; Moon & Kirton, 2016). The jugal has a long suborbital bar and posteriorly ascending process. The surborbital bar is straight and narrows dorsoventrally anteriorly, differing from the uniform jugal seen in Athabascasaurus bitumineus (Druckenmiller & Maxwell, 2010). The suborbital bar is flattened in cross-section and not sub-circular as in O. icenicus and L. stellans (Fischer et al., 2013; Moon & Kirton, 2016). It lacks the interdigitating anterior end of Brachypterygius extremus for insertion with the premaxilla (Kirton, 1983). In PMO 222.669, the medial side has an anteroposteriorly elongated depression for articulation with the maxilla and a second, smaller depression situated more posteriorly. Similar to the holotype of P. hoybergeti, the jugal lacks a distinct posteroventral heel such as that seen in O. icenicus and Platypterygius australis (Kear, 2005; Moon & Kirton, 2016), and instead curves gradually into the posterior process, which ascends dorsally at an angle of approximately 60° relative to the suborbital bar. In lateral view, it is widest anteroposteriorly at the dorsoventral midpoint, above which it narrows dorsally, in contrast to the P. hoybergeti holotype. The size and outline of the posterior portion varies intraspecifically in O. icenicus (e.g., CAMSM J29861 vs. NHMUK PV R8653, L.L. Delsett, personal observation; Moon & Kirton, 2016), thus, the difference to the holotype might represent individual variation. The posterodorsal tip has a dorsoventrally oriented depression medially, a feature also found in the holotype of P. hoybergeti.

Lacrimal (Fig. 8C)

The left lacrimal was found in the left supratemporal fenestra. It is similar to that in the holotype (SVB 1451) (Druckenmiller et al., 2012), but because it is disarticulated, additional details can be added. The dorsalmost portion of the dorsal projection shows an interdigitating structure with the prefrontal as in Platypterygius australis, O. icenicus, and Simbirskiasaurus birjukovi (Kear, 2005; Fischer et al., 2014a; Moon & Kirton, 2016), and some of the projections of the prefrontal are preserved. The dorsal projection and middle part of the element are anteroposteriorly narrower than in Paraophthalmosaurus (UPM EP-II.7(1235) L.L. Delsett, personal observation) (Arkhangelsky, 1997) and Simbirskiasaurus birjukovi (Fischer et al., 2014a). On the lateral surface, fine striations radiate from the dorsal end of the narrow margin and in all directions. The medial surface is flatter than the lateral. The middle portion of the element and the dorsal projection are rugose and bears at least one foramen, as in Acamptonectes densus and Platypterygius australis (Kear, 2005; Fischer et al., 2012). The anterior process has a further anterior reach than in Aegirosaurus leptospondylus and Caypullisaurus bonapartei (Bardet & Fernández, 2000; Fernández, 2007). In ventral view, the anterior process has a ridge running in anteroposterior direction in the lateral part and three small foramina medial to this. The posterior projection is divided into two anteroposteriorly directed grooves that decrease in depth posteriorly, for articulation with the maxilla and the jugal (Kear, 2005; Moon & Kirton, 2016).

Vomer (Fig. 8D)

The left vomer was disarticulated close to the left pterygoid on the left side of the skull and is incomplete. Anteriorly, it is elongate and mediolaterally and dorsoventrally narrow; in the posterior half, the element widens into a dorsoventrally oriented sheet that gradually decreases in height posteriorly. The dorsal sheet has an uneven outline in lateral view with at least two broken processes in the dorsal portion oriented dorsally and anteriorly, as in O. icenicus and Gengasaurus nicosiai (Moon & Kirton, 2016; Paparella et al., 2016). The element is laterally convex with a lateral side that is more uneven than the smoother, medial side. The posterior extension is mediolaterally and dorsoventrally narrow and longitudinally striated. Ventrally, just posterior to the maximum dorsoventral height, there are striations radiating in all directions, which is also true for the medial and lateral surfaces.

Pterygoid (Fig. 8E)

In PMO 222.669, the left pterygoid is complete except for the very tips of the thin lateral and dorsal processes, whereas the right preserves a portion of the posterior ramus. In the holotype of P. hoybergeti (SVB 1451), the pterygoid is only partially known (Druckenmiller et al., 2012). Overall the element is similar to the exposed portion seen in the holotype. Similar to other ophthalmosaurids as well as the non-ophthalmosaurid Hauffiopteryx, the pterygoid has a sheet-like anterior palatal ramus and a more complex, posterior quadrate ramus with three processes (Kear, 2005; Fischer et al., 2011; Marek et al., 2015; Moon & Kirton, 2016). The anterior portion is straight and not medially curved as in Platypterygius australis (Kear, 2005). Posterior to this is a dorsoventrally oriented ridge that lacks the medial terrace seen in O. icenicus (Moon & Kirton, 2016) (MANCH L10307, A.J. Roberts, personal observation). Posterior to the ridge is a mediolaterally wide horizontal shelf, which is very thin and not completely preserved, but with some anterior processes as in O. icenicus (Moon & Kirton, 2016). Anterior to the quadrate ramus, the horizontal shelf is constricted, as in O. icenicus and the P. hoybergeti holotype (Druckenmiller et al., 2012; Moon & Kirton, 2016). The ventral side of the horizontal shelf is concave. On the quadrate ramus, the lateral and medial portions are anteroposteriorly narrower than in O. icenicus and Platypterygius australis (Kear, 2005; Moon & Kirton, 2016) and of approximately the same size, unlike Sveltonectes insolitus where the lateral wing is smaller (Fischer et al., 2011). The dorsal portion has a more robust base than in O. icenicus (Moon & Kirton, 2016), with a square outline. Sisteronia seeleyi also has a dorsal process with a thick base (Fischer et al., 2014b). The area between the dorsal and lateral process articulated with the quadrate. In posterior view, the ventral side of the posterior ramus is concave as in Sveltonectes insolitus (Fischer et al., 2011), but differs in lacking a forked posterior end.

Hyoid (Fig. 8F)

A complete hyoid is preserved in PMO 222.669, but this element is not known in the holotype specimen of P. hoybergeti (SVB 1451) (Druckenmiller et al., 2012). It was found near the right side of the skull and has been displaced; as such it is not possible to definitively orient the element (Kear, 2005; Kolb & Sander, 2009; Fischer et al., 2011). The hyoid is transversely compressed as in O. icenicus (Moon & Kirton, 2016) and more curved than that seen in J. lundi (Roberts et al., 2014) and one O. icenicus specimen (NHMUK R3013, A.J. Roberts, personal observation). The hyoid is oval in cross-sectional shape, differing from that of Platypterygius hercynicus and Sveltonectes insolitus, which are circular in cross-section for parts of its length (Kolb & Sander, 2009; Fischer et al., 2011). In medial and lateral view, the dorsoventral height of the element is constant for almost the entire length, but widens dorsoventrally into a spatula-shaped wider end. The other end has a square outline in medial view as in Platypterygius hercynicus (Kolb & Sander, 2009) and is rugose, probably for attachment of soft tissue. Both ends have elongate depressions on the medial and lateral sides.

Supratemporal (Fig. 7)

Both supratemporals are preserved (Fig. 7). As in O. icenicus, the most robust portion of the element makes up the posterolateral corner of the cranium. From the posterolateral portion, the anterior ramus extends on the lateral side of the supratemporal fenestra and meets with the postfrontal anteriorly. The supratemporal does not contact the frontal in contrast to Platypterygius hercynicus (Kolb & Sander, 2009). In posterior view, one distinct and possibly two posterolateral tubercles can be seen, possibly for muscle attachment. Two tubercles are described for O. icenicus and UAMES 3411 (Druckenmiller & Maxwell, 2013; Moon & Kirton, 2016). Two rami descend on the posterior side of the skull, one being directed laterally and one medially, diagonally toward the middle of the basicranium. The quadrate articulates with the lateral side of the lateral ramus. The lateral process also contacts the stapes, as in L. stellans, O. icenicus, and Baptanodon natans (Gilmore, 1905; Fischer et al., 2013; Moon & Kirton, 2016). The medial ramus overlaps the opisthotic and terminates approximately on the middle of this element. The medial portion of the element is thickened and becomes narrower medially.

Basioccipital (Figs. 9A–9D)

As in the P. hoybergeti holotype (SVB 1451), the basioccipital lacks exposure of extracondylar area in posterior view (Fig. 9A). The extracondylar area is represented by a very shallow peripheral groove of finished bone laterally and ventrally (Figs. 9C–9D) (Druckenmiller et al., 2012; Fischer et al., 2014a) in contrast to many ophthalmosaurine ophthalmosaurids with a large part of the extracondylar visible in posterior view, such as O. icenicus (Moon & Kirton, 2016), but is instead largely similar to Grendelius alekseevi (Zverkov, Arkhangelsky & Stenshin, 2015). The dorsal surface is better preserved in the new specimen (Fig. 9B). The floor of the foramen magnum is wide posteriorly and decreases in mediolateral width anteriorly, and has slightly elevated lateral walls. The floor of the foramen magnum does not form a dorsal process as in Sisteronia seeleyi (Fischer et al., 2014b). The exoccipital facets occupy most of the area lateral to the notochordal groove. The opisthotic facets are situated on the dorsal portion of the lateral surface, and are dorsoventrally elongated. These facets are raised as in Sveltonectes insolitus in contrast to Acamptonectes densus (Fischer et al., 2011, 2012), where they are small. As in the holotype, a ventral notch is absent, but a small oval depression is present on the ventral surface close to the anterior margin (Fig. 9D).

Basisphenoid (Fig. 9E)

The parabasisphenoid is visible in ventral view only as it is preserved in articulation to the skull roof. The parasphenoid is preserved, but its anterior portion is not accessible. It originates in the anterior portion of the ventral surface as in other ophthalmosaurids (Kear, 2005; Moon & Kirton, 2016). It is oval in cross-section and dorsoventrally flattened. The basipterygoid processes originate in the anteriormost portion of the ventral surface of the basisphenoid and are laterally directed. They are distinct, but smaller than in Platypterygius australis (Kear, 2005) and Sisteronia seeleyi (Fischer et al., 2014b) and instead resemble O. icenicus (Moon & Kirton, 2016) and Sveltonectes insolitus (Fischer et al., 2011). In the anteroposteriorly midpoint of the ventral surface is an anteroposteriorly oriented oval depression as in Arthropterygius chrisorum (Maxwell, 2010). The carotid foramen exits on the posterior surface as in Arthropterygius chrisorum (Maxwell, 2010), the Russian Arthropterygius sp. specimen SGM 1502 and Platypterygius australis (posteroventral) (Kear, 2005; Zverkov et al., 2015), in contrast to the ventral exit in, for example, O. icenicus. Grendelius alekseevi and Sveltonectes insolitus (Fischer et al., 2011; Zverkov, Arkhangelsky & Stenshin, 2015; Moon & Kirton, 2016). The foramen is positioned in the ventralmost portion of the posterior surface so that it makes an incision to the posterior margin of the ventral surface in ventral view. The margin is very similar to that in Brachypterygius extremus, but in that species the carotid is interpreted to exit ventrally (McGowan, 1976; Kirton, 1983). The element is mediolaterally narrowest in the posterior portion as is also seen in many ophthalmosaurids, for example, Platypterygius australis and Brachypterygius extremus (McGowan, 1976; Kear, 2005).

Exoccipital (Fig. 9B)

The exoccipitals are better preserved in an articulated state in the new specimen than in the holotype (SVB 1451) (Druckenmiller et al., 2012). Two foramina perforate the ventral portion of the element in PMO 222.669; with the posterior being the largest, similar to those in O. icenicus and Athabascasaurus bitumineus (Druckenmiller & Maxwell, 2010; Moon & Kirton, 2016). The holotype (SVB 1451) of P. hoybergeti lacks foramina (Druckenmiller et al., 2012), probably due to taphonomic distortion or individual variation (Maisch, 1997). Only one foramen is found in Sveltonectes insolitus whereas three are present in Platypterygius australis and Brachypterygius extremus (McGowan, 1997; Kear, 2005; Fischer et al., 2011). The ventral surface is convex and finely pitted, probably for cartilage in the articulation with the basioccipital. The exoccipitals had a small contact with the opisthotic laterally as in O. icenicus (Moon & Kirton, 2016).

Opisthotic (Fig. 9F)

In PMO 222.669, both opisthotics are preserved in articulation with the basioccipital and supratemporals, although the anterior surfaces are not visible. Information regarding these elements is not available in the holotype of P. hoybergeti (SVB 1451) due to poor preservation. In posterior view, the element is pentagonal with a straight ventral margin. The incomplete medial margin is the longest. The element possesses a small, medially directed process probably for articulation with the exoccipital. This morphology differs from Acamptonectes densus, which lacks this process and instead has a concave medial margin (Fischer et al., 2012). The paroccipital process has a finished surface and is dorsolaterally directed to articulate with the supratemporal. Compared to O. icenicus and Platypterygius australis, the paraoccipital process is short and to a small degree set off from the rest of the element (Kear, 2005; Moon & Kirton, 2016), being in this regard more similar to that of Sisteronia seeleyi (Fischer et al., 2014b) and more basal ichthyosaurs and the early Jurassic Hauffiopteryx (Marek et al., 2015). The element lacks the lateral ridge present in Acamptonectes densus (Fischer et al., 2012). The posterior surface of the opisthotic is flat and only lightly pitted. The ventral side has a shallow groove dividing the facet for articulation with the stapes into two parts.

Stapes (Fig. 9G)

Both stapes are preserved nearly in articulation with the basicranium. The most complete stapes from the holotype (SVB 1451) was originally interpreted to be a left as it was found disarticulated on the left side of the skull (Druckenmiller et al., 2012). However, based on the articulated material from the new specimen, we reinterpret that from the holotype as a right element that was displaced to the left side of the skull. Morphologically, the element is overall very similar to that of the holotype with a gracile shaft possessing an oval cross-section. The lateral head is only very slightly anteroposteriorly wider compared to the shaft, slightly less than in the holotype (Druckenmiller et al., 2012) and similar to J. lundi (Roberts et al., 2014) in contrast to the expanded lateral head in, for example, Grendelius alekseevi (Zverkov, Arkhangelsky & Stenshin, 2015). In anterior view, the quadrate facet covers most of the lateral head of the element. The medial head is much wider than the shaft and takes up approximately half of the mediolateral length of the element. In dorsal view, the opisthotic facet is triangular and more posteriorly directed than in O. icenicus (Moon & Kirton, 2016). In ventral view, the basioccipital and basisphenoid facets are barely visible, but in medial view, they are offset at an angle of approximately 45° and lack a ridge between them, as observed in Platypterygius australis (Kear, 2005). L. stellans has a large hyoid process, whereas it is lacking in Sisteronia seeleyi, and this feature is probably variable in ophthalmosaurids (Fischer et al., 2012, 2013). In PMO 222.669, this process is ventrally directed.

Quadrate (Figs. 9H–9I)

The right quadrate is distorted whereas the left is very well preserved. In the holotype of P. hoybergeti (SVB 1451), the quadrates were not described due to poor preservation (Druckenmiller et al., 2012). The quadrate has the classic C-shape with a convex medial outline of the pterygoid lamella, and a laterally expanded occipital lamella, such as that seen in O. icenicus and several other ophthalmosaurids (Fischer et al., 2013; Moon & Kirton, 2016). In dorsal view, the occipital lamella forms an angle of 120–130° relative to the pterygoid lamella, similar to Acamptonectes densus (Fischer et al., 2012). The element is more laterally expanded than Platypterygius australis (Kear, 2005), Grendelius alekseevi (Zverkov, Arkhangelsky & Stenshin, 2015), and UAMES 3411 (Ophthalmosauridae indet.; (Druckenmiller & Maxwell, 2013). In contrast, Platypterygius hercynicus lacks an occipital lamella (Kolb & Sander, 2009). The facet for the pterygoid covers almost the entire lamella. As in most ophthalmosaurids, a deep stapedial facet is situated just dorsal to the dorsal surface of the articular condyle (Fig. 9H), although it is more ventrally placed in Sisteronia seeleyi and Grendelius alekseevi (YKM 56702) (Fischer et al., 2014b; Zverkov, Arkhangelsky & Stenshin, 2015; Moon & Kirton, 2016). The ventral margin of the facet is thickened, similar to O. icenicus and Acamptonectes densus (Fischer et al., 2012; Moon & Kirton, 2016). Medial to the stapedial facet is a tiny foramen. The medial margin is pitted and is anteroposteriorly narrow dorsally and widens ventrally. Platypterygius australis differs in having a groove along this margin (Kear, 2005) whereas the ophthalmosaurid UAMES 3411 is similar to PMO 222.669 (Druckenmiller & Maxwell, 2013). In ventral view the articular condyle is roughly triangular (Fig. 9I). On the posterior side is a larger facet for the articular, which is typical in most ophthalmosaurids (Kear, 2005; Fischer et al., 2012), but it is the smaller in Gengasaurus nicosiai (Paparella et al., 2016). On the lateral margin is a smaller facet for articulation with the surangular. A small facet lateral and posterior to this is interpreted to be for the quadratojugal.

Articular (Figs. 10A–10C)

Both articulars are preserved, the left being found in articulation with the surangular and covered medially by the posterior bar of the prearticular, similar to O. icenicus (Moon & Kirton, 2016). The articulars of PMO 222.669 are similar to those of the holotype (SVB 1451), but better preservation has revealed new information. As in the holotype, the articular differs from most other ophthalmosaurids in being dorsoventrally tallest at the anterior end and broader posteriorly. In medial view, ophthalmosaurid articulars are most commonly oval (Kear, 2005; Maxwell, 2010; Fischer et al., 2014b; Moon & Kirton, 2016). In PMO 222.669, the articular is rectangular with a slight constriction at the anteroposterior midpoint as in the holotype of P. hoybergeti. The medial side (Fig. 10A) is convex and has a ridge running in the anteroposterior direction which is not as narrow as in Baptanodon natans (Gilmore, 1905). The ventral flange bears an elongated facet posteroventrally. A facet on the dorsal portion of the medial surface has a roughened surface, but the dorsal surface is not thickened as in Platypterygius australis (Kear, 2005). The element widens slightly mediolaterally at the posterior end, which has a rounded outline in medial or lateral view. The surface at the posterior end is more rugose than the rest of the medial side. The lateral surface (Fig. 10B) has an elevated area in its anteroventral region and a double diagonal ridge crossing the element, as in Arthropterygius chrisorum (Maxwell, 2010). In contrast, this ridge runs horizontally in O. icenicus (Moon & Kirton, 2016). The area dorsal to this ridge articulated with the surangular.

Dentary (Fig. 6)

In PMO 222.669, both dentaries are preserved and the left can be studied in both lateral and medial views, whereas they are known in lateral view only from the holotype. The anterior tip is narrow in lateral view compared to the blunter and wider tip in Brachypterygius and Acamptonectes densus (McGowan, 1976; Fischer et al., 2012) and has a ventral margin that curves dorsally, as in Platypterygius australis (Kear, 2005). Overall, the dentary is more robust than the slender one in Aegirosaurus leptospondylus (Bardet & Fernández, 2000). Anteriorly in the element are preserved several teeth that are smaller than the others and appear to be close to life position. Posteriorly, the dental groove widens mediolaterally, resulting in a wide and very shallow groove in the posteriormost half of the elements. Distinct alveoli are absent. On the lateral side, some large foramina are present in the anteriormost portion of the ramus, organized into two rows. These develop into a groove posteriorly that continues for the majority of the anteroposterior length of the element and that deflect ventrally in the most posterior region, possibly for articulation with the angular. The posteriormost portion of the lateral side bears a tall anteroposteriorly oriented depression that widens posteriorly, probably for articulation with the surangular. The medial surface is concave as in O. icenicus and Platypterygius australis (Kear, 2005; Moon & Kirton, 2016) through the entire anteroposterior length of the element, mainly in the ventral half.

Splenial (Fig. 6)

Both splenials are completely preserved. In the holotype (SVB 1451), the left splenial is partly visible in ventral view of the skull. The anterior end is forked as in O. icenicus and Platypterygius australis (Kear, 2005; Moon & Kirton, 2016), but the fork is relatively longer than in these taxa and with very long anterior finger-like projections. As in O. icenicus, the ventral finger is laterally concave and significantly longer than the dorsal (Moon & Kirton, 2016). The main body of the element (excluding the forked anterior portion) is 5–10 cm anteroposteriorly shorter than the surangular and the dentary. The dorsoventral height gradually decreases posteriorly to approximately the same height as the surangular. For the posteriormost centimeters, the dorsal margin abruptly slopes posteriorly and has an uneven outline as in the holotype specimen of P. hoybergeti and Platypterygius australis (Kear, 2005). The posterior margin is significantly thinner than the thickened and subcircular shape found in Pervushovisaurus (Fischer et al., 2014a), and from Cryopterygius kristiansenae, which has a forked posterior end (Druckenmiller et al., 2012). The ventral margin is concave in posterior view as in Platypterygius australis (Kear, 2005). In lateral view, at a point near of the anterior one third of its length, is a small area on the ventral surface where the striations in the surface radiates from in all directions, which might correspond to the point in Muiscasaurus where the splenial becomes less robust because the symphysis ends (Maxwell et al., 2015). The element is longitudinally striated on both its medial and lateral sides posterior to this point. Posteriorly, the lateral surface facing the surangular has two anteroposteriorly oriented ridges.

Surangular (Figs. 10D–10F)

Both surangulars are preserved. The right surangular is complete in anteroposterior length (Fig. 10D), whereas the left is three dimensional, but lacks its anterior tip (Fig. 10E). Anteriorly, the surangular curves dorsally in lateral view and is overall similar to O. icenicus (Moon & Kirton, 2016). The dorsal margin is mediolaterally wider and more rounded than the ventral, giving an inverted tear-shape cross-section. In contrast, Pervushovisaurus bannovkensis has a T-shaped cross-section (Fischer et al., 2014a). The anterior tip is mediolaterally and dorsoventrally narrow, and gradually widens posteriorly. In medial view is an anteriorly placed groove midway between the dorsal and ventral margins, interpreted as the symphyseal portion of the Meckelian canal (Moon & Kirton, 2016). The groove disappears just anterior to the anteroposterior midpoint of the element and is not visible posteriorly, unlike Gengasaurus nicosiai (Paparella et al., 2016) and Platypterygius australis (Kear, 2005). In lateral view (Fig. 10D), the longitudinal fossa surangularis is prominent, as in O. icenicus (Moon & Kirton, 2016), but unlike Sveltonectes insolitus, which lacks this feature (Fischer et al., 2011). The fossa surangularis is grooved with a small dorsal overhang, and is dorsoventrally deepest in the anteroposterior midpoint of the element. The surangular foramen is placed midway between the dorsal and ventral margins where the fossa surangularis ends posteriorly. The left surangular has two foramina on the lateral side. Posterior to the two foramina are two dorsal processes, the smallest and most anterior of which is interpreted as the paracoronoid process. Posterior to this is a dorsomedially directed preglenoid process, also referred to as M. adductor mandibulae extremus (MAME) process. This is similar to O. icenicus (Moon & Kirton, 2016) and J. lundi (Roberts et al., 2014); in contrast Platypterygius australis has only one process (Kear, 2005). The presence of a MAME process is intraspecifically variable in Acamptonectes densus (Fischer et al., 2012). The posteriormost portion of the surangular is shifted laterally posterior to the excavation for the glenoid fossa. The posterior margin has a square outline in lateral or medial view and resembles Platypterygius hercynicus (Kolb & Sander, 2009) and P. australis (Kear, 2005). It is laterally convex and is more coarsely striated than the rest of the element.

Angular (Figs. 10G–10H)

Both angulars are preserved, disarticulated, and displaced. It is not known how much of the mandible the angular covered when in articulation. However, the element appears shorter relative to the surangular than in Aegirosaurus leptospondylus, O. icenicus, and Cryopterygius kristiansenae, and it lacks the long, anterior extension seen in O. icenicus (Bardet & Fernández, 2000; Druckenmiller et al., 2012; Moon & Kirton, 2016). Anteriorly, the element is instead made up of two thin flanges. The smaller flange of the two is situated medially and the other flange is dorsoventrally taller and on the lateral side. This is more similar to Platypterygius australis, but differs in being longer anteriorly compared to PMO 222.669 (Kear, 2005). At the anteriormost end of the angular of PMO 222.669 is a narrow opening between two flanges ventrally. The shape and size of the two flanges resulted in more lateral than medial exposure of the angular. O. icenicus, Pervushovisaurus bannovkensis, and Platypterygius australis have two or three grooves on the dorsal surface of the angular, and some specimens of Acamptonectes densus have only one groove, as in PMO 222.669 (Kear, 2005; Fischer et al., 2012, 2014a; Moon & Kirton, 2016). The element curves dorsally in the posteriormost portion and has an oval cross-section.

Prearticular (Figs. 10E–10F)

Both prearticulars are preserved, but neither of them is complete anteriorly. The left prearticular was preserved in articulation with the surangular and articular. In life, this element was probably completely covered in medial view by the (often) disarticulated splenial, similar to O. icenicus (Moon & Kirton, 2016) and Platypterygius australis (Kear, 2005). The element is mediolaterally thin. The maximum dorsoventral height is situated just anterior to the paracoronoid process on the surangular, similar to O. icenicus and J. lundi (Roberts et al., 2014; Moon & Kirton, 2016). The dorsal margin is relatively taller compared to the rest of the element than in Platypterygius australis (Kear, 2005). Between the dorsal and ventral margin a small foramen pierces the element. Striations spread out from this point in all directions on the medial and lateral faces of the element. The element decreases abruptly in height toward the posterior bar, which is dorsoventrally short and more coarsely striated than the rest of the element, possibly for muscle attachment (Kear, 2005). The prearticular extends posteriorly almost to the posterior end of the articular. The ventral margin is longitudinally grooved and bears coarse striations in all directions.

Dentition (Figs. 6C–6F)

At least 126 teeth are preserved in the anteriormost part of the rostrum, displaced from the dental groove of the premaxillae and dentaries. Additional disarticulated teeth are distributed over the rest of the skull. In the anteriormost part of one dentary are preserved some small teeth almost in life position, similar to the holotype specimen, although this was not noted in the original description (Druckenmiller et al., 2012). The majority of the teeth are between 20 and 34 mm in height including the crown and root, and crown height varies from 12 to 15 mm in most teeth. The teeth are not distinctly different in shape nor size from those in the holotype (SVB 1451) (Druckenmiller et al., 2012). The crown covers between one third and one half of total tooth height, more than in Sveltonectes insolitus (Fischer et al., 2011). As in Cryopterygius kristiansenae, the crown is slightly distally curved and has a subtly ridged enamel except in the uppermost portion where they are smooth (Druckenmiller et al., 2012). The base of the enamel layer is straight and well-defined and the crowns are circular in cross-section. An acellular cementum ring (Maxwell, Caldwell & Lamoureux, 2011) just ventral to the enamel border transitions into fine striations ventrally as in the Russian Arthropterygius sp. (Zverkov et al., 2015). The root is quadrangular in cross-section and slightly compressed in what is probably the labiolingual plane; in this regard it is more similar to Sisteronia seeleyi than to O. icenicus and the thickened root in Paraophthalmosaurus (“Yasykovia”) (Arkhangelsky, 1997; Efimov, 1999b; Fischer et al., 2014b; Moon & Kirton, 2016). Many roots show distinct resorption cavities.

Sclerotic plates

At least 13 disarticulated sclerotic plates are preserved, but many details cannot be discerned. The sclerotic plates are relatively longer compared to width than Platypterygius hercynicus (Kolb & Sander, 2009), being more similar to O. icenicus (Moon & Kirton, 2016). None of the plates has any thickened portion. The most complete plate measures 62 mm in length, which is similar to J. lundi (Roberts et al., 2014), 37 in maximum width (outer edge), 21 mm minimum (inner edge).

Pectoral girdle

Clavicle-interclavicle complex (Fig. 11A)

As in O. icenicus, the medial stem of the T-shaped interclavicle is longer than the transverse bar (Moon & Kirton, 2016), in contrast to Caypullisaurus bonapartei, where the transverse bar is twice the length of the medial stem (Fernández, 1997) and Aegirosaurus leptospondylus where the two are approximately the same length (Bardet & Fernández, 2000), L.L. Delsett, personal observation). The medial stem is unusually mediolaterally wide compared to many other ophthalmosaurids (Druckenmiller et al., 2012) and resembles Grendelius alekseevi (Zverkov, Arkhangelsky & Stenshin, 2015), but this might be due to compression. The stem narrows slightly toward its posterior end. The visceral side is flat, and in contrast to J. lundi, there appears to be no trough (Roberts et al., 2014). The medial portion of the clavicle is anteroposteriorly wider than in Athabascasaurus bitumineus (Druckenmiller & Maxwell, 2010), and is more similar to Paraophthalmosaurus (UPM EP-II-7(1235); L.L. Delsett, personal observation). The anterior edge is thickened, and the facets for the scapulae are elongated. The posterodorsal tip of the clavicle is dorsoventrally narrow, and widens gradually anteriorly, resembling PMO 222.667, in contrast to the clavicle of J. lundi, which widens more abruptly anteriorly (Roberts et al., 2014).

Scapula (Figs. 11B–11F)

The left scapula has suffered some deformation in the anterior portion, affecting the shape of the acromion process, which has become flattened, and the description is mainly based on the right element. The anterior portion of the scapula is expanded more ventrally than the dorsal height of the acromion process (Figs. 11C and 11E). This is similar to that of O. icenicus and Grendelius alekseevi (Zverkov, Arkhangelsky & Stenshin, 2015; Moon & Kirton, 2016), whereas Platypterygius americanus and P. australis, in contrast, have an anterior part almost similarly expanded dorsally and ventrally (Maxwell & Kear, 2010; Zammit, Norris & Kear, 2010). In lateral view (Fig. 11C), the acromion process is only slightly expanded dorsally, less than PMO 222.667 and Paraophthalmosaurus (“Yasykovia”; UPM EP-II-7(1235), L.L. Delsett, personal observation) (Efimov, 1999b) and more similar to Cryopterygius kristiansenae and Grendelius alekseevi (Druckenmiller et al., 2012; Zverkov, Arkhangelsky & Stenshin, 2015). It has a large dorsolateral flange approaching the size of Sveltonectes insolitus (Fischer et al., 2011) and PMO 224.250, being considerably larger than in PMO 222.667. In anterior view, there is a mediolaterally narrow portion ventral to the acromion process, but lacks a notch as in Sveltonectes insolitus (Fischer et al., 2011). Ventral of this is the mediolaterally expanded articular region with the coracoid and glenoid facets, the latter being the smaller of the two. Both facets are deeply rugose and more demarcated from each other than in Keilhauia nui, PMO 222.667 (Delsett et al., 2017) and Platypterygius americanus (Maxwell & Kear, 2010). The glenoid facet is almost circular in shape and faces more ventrally than in PMO 222.667. The dorsal margin of the posterior shaft is nearly straight whereas the ventral margin is concave resulting in a slight dorsoventral expansion in the posterior end as in Caypullisaurus bonapartei (Fernández, 1997, MLP 83-XI-16-1, A.J. Roberts, personal observation), somewhat more than K. nui (Delsett et al., 2017) and Baptanodon natans (Gilmore, 1905), but less than in Sveltonectes insolitus (Fischer et al., 2011), Platypterygius americanus (Maxwell & Kear, 2010), and Grendelius alekseevi (Zverkov, Arkhangelsky & Stenshin, 2015). The posterior shaft is relatively short compared to the anterior portion compared to the more slender shaft in Grendelius alekseevi (Zverkov, Arkhangelsky & Stenshin, 2015).

Coracoid (Figs. 11G–11H)

The coracoid is slightly anteroposteriorly longer than mediolaterally wide, as in Cryopterygius kristiansenae (Druckenmiller et al., 2012) and Undorosaurus spp. (Efimov, 1999a) but not as narrow as Paraophthalmosaurus (UPM EP-II-7(1235), L.L. Delsett, personal observation) and N. enthekiodon (Kirton, 1983). The scapular and glenoid facets are angled medially so that the coracoid is mediolaterally narrower in the posterior part than in the middle, and makes the outline of the coracoid resemble that of Sveltonectes insolitus (Fischer et al., 2011), differing from the hexagonal coracoid in Acamptonectes densus (Fischer et al., 2012) and the more circular element in Platypterygius hercynicus (Kolb & Sander, 2009) and P. australis (Zammit, Norris & Kear, 2010).

The coracoid possesses a well-defined anterior notch and an anteromedial process. Relative to the size of the anteromedial process, the anterior notch is mediolaterally narrow compared to O. icenicus and Acamptonectes densus (Fischer et al., 2012; Moon & Kirton, 2016). The anterior notch is anteroposteriorly shorter than K. nui (Delsett et al., 2017) but longer and narrower than in PMO 222.667 and Arthropterygius chrisorum (Maxwell, 2010). The anterior process is straighter in dorsal view than the processes found in Sveltonectes insolitus (Fischer et al., 2011) and J. lundi (Roberts et al., 2014). The dorsal surface is flat, which is similar to PMO 222.667, in contrast to O. icenicus, which has coracoids that have convex (“saddle-shaped”) dorsal and ventral sides (Moon & Kirton, 2016). The posterior margin is dorsoventrally thin. The intercoracoid facet is not visible due to either taphonomic or osteological damage. The scapular and glenoid facets are well demarcated and the glenoid facet is the longest. The scapular facet is relatively longer compared to the glenoid facet than in O. icenicus and Grendelius alekseevi (Zverkov, Arkhangelsky & Stenshin, 2015; Moon & Kirton, 2016) and PMO 222.669 resembles Sveltonectes insolitus (Fischer et al., 2011) in this aspect. The articular surface of the glenoid facet faces ventrolaterally.

Humerus (Fig. 12)

The humerus is oriented based on McGowan & Motani (2003). The proximal end is slightly anteroposteriorly narrower than the distal end, similar to but less pronounced than in J. lundi (Roberts et al., 2014). Sveltonectes insolitus (Fischer et al., 2011) and Platypterygius americanus possess a wider proximal than distal end (Maxwell & Kear, 2010). In dorsal view, the proximal end is not straight as in J. lundi and Arthropterygius chrisorum (Maxwell, 2010; Roberts et al., 2014) but is strongly convex.

The dorsal process of the humerus of PMO 222.669 originates near the midpoint of the proximal end similar to that seen in the Indian Ophthalmosauridae indet. KGMV-0501 (Prasad et al., 2017), and its long axis is angled less anteriorly than in PMO 222.667, PMO 224.250, and PMO 222.658. The dorsal process extends to the proximodistal midpoint, as in Undorosaurus gorodischensis (Efimov, 1999a) and O. icenicus (Moon & Kirton, 2016). This is proximodistally shorter than in Arthropterygius chrisorum (Maxwell, 2010) and Platypterygius hercynicus (Kolb & Sander, 2009), but longer than in K. nui (Delsett et al., 2017). In anterior view the deltopectoral crest is dorsoventrally short and does not reach the midpoint of the element on the ventral side, similar to K. nui (Delsett et al., 2017) and Arthropterygius chrisorum (Maxwell, 2010). This contrasts the triangular and protruding crest found in many other ophthalmosaurids (e.g., PMO 222.658, PMO 222.667, PMO 224.250, (Fischer et al., 2012)). Its placement along the anterior margin of the ventral side is similar to O. icenicus (Moon & Kirton, 2016). Two pairs of rugosities, possibly representing muscle insertion points, are located on the dorsal and ventral sides in the distal part of the element. Small processes in this portion of the humerus are also found in Sveltonectes insolitus and Ichthyosaurus anningae and might represent the same structures (Fischer et al., 2011; Lomax & Massare, 2015).

There are three distal articular facets for a preaxial accessory element, and the radius and ulna. This is similar to all other SML ophthalmosaurids (including the partial humerus of the P. hoybergeti holotype) with the exception of Cryopterygius kristiansenae, which has two facets on one humerus and three facets on the other in addition to several other ophthalmosaurids, for example, Caypullisaurus bonapartei, Acamptonectes densus, and Undorosaurus (Fernández, 1997; Efimov, 1999a; Druckenmiller et al., 2012; Fischer et al., 2012; Arkhangelsky & Zverkov, 2014; Roberts et al., 2014; Zverkov et al., 2015; Delsett et al., 2017). In contrast, Aegirosaurus leptospondylus, Maiaspondylus lindoei, Grendelius, and Brachypterygius extremus have a third facet for the intermedium (McGowan, 1976; Bardet & Fernández, 2000; Maxwell & Caldwell, 2006; Zverkov, Arkhangelsky & Stenshin, 2015), whereas Platypterygius americanus has a third facet for a postaxial accessory element (Maxwell & Kear, 2010). In PMO 222.669, the facet for the radius is only slightly larger than the ulnar facet. In O. icenicus specimens, the three facets vary in relative size and orientation (Moon & Kirton, 2016). As in Gengasaurus nicosiai (Paparella et al., 2016) and Arthropterygius chrisorum (Maxwell, 2010) the ulnar facet is not posteriorly deflected, unlike in K. nui (Delsett et al., 2017).

The right humerus may bear a pathological feature on the ventral side near the posterior margin, probably a rugosity due to healing, resulting from a simple trauma (Pardo-Pérez et al., 2017).

Zeugopodium and autopodium (Figs. 12A, 12B and 12D)

Because the preserved forefin elements remain in articulation, it is possible to assign them with confidence, based on O. icenicus and Cryopterygius kristiansenae (Druckenmiller et al., 2012; Moon & Kirton, 2016). The total number of digits is unknown. The two larger elements that articulate with the distal end of the humerus are recognized as the radius and ulna, leaving the anteroposteriorly smaller and anterior element to be interpreted as the preaxial accessory element. The preaxial accessory element articulates with the humerus, radius, radiale and a small unassigned distal element, possibly part of a supernumerary digit, and has no additional anterior facets. The element is marginally anteroposteriorly longer than proximodistally wide and is less dorsoventrally tall proximally than the radius and ulna.

The radius articulates with the intermedium and radiale distally, the ulna posteriorly and the preaxial accessory element anteriorly. Of these, the radiale facet is the longest, followed by the intermedium, preaxial accessory, and ulna facet, respectively. The humeral facet resembles that of the ulna in being convex, but not as extreme as in Arthropterygius chrisorum (Maxwell, 2010).

The ulna is anteroposteriorly wider than proximodistally long, and proximodistally longer than the radius as in J. lundi (Roberts et al., 2014) and Platypterygius hercynicus (Kolb & Sander, 2009). The proximal end is dorsoventrally approximately twice as tall as the distal end. The articular facet with the humerus is convex. In addition, the ulna articulates with the radius, intermedium and ulnare; the facet for the radius is approximately half the length of that for the intermedium and ulnare, which are subequal in length. The posterior margin of the ulna is the dorsoventrally shortest of the margins so that the ulna element tapers posteriorly (Fischer et al., 2012), and it bears two distinct surfaces. One is situated proximally and is approximately one cm long, dorsoventrally thin and straight in dorsal view, with a finished surface, and is not concave as in Acamptonectes densus and O. icenicus (Fischer et al., 2012). The other surface is a small, fifth facet directed posteriorly and slightly distally, probably for contact with a pisiform as in Acamptonectes densus (Fischer et al., 2012).

The radiale, intermedium and ulnare are all anteroposteriorly wider than proximodistally long. The ulnare articulates with the intermedium anteriorly, an unidentified distal element that could represent metacarpal 5 and distal carpal 4, with the facet for the ulna being the longest. The intermedium is anteroposteriorly wider than proximodistally long and articulates with the ulnare, distal carpal 4 and 3 and possibly the radiale. The radiale articulates with distal carpals 2 and 3, as well as the unidentified anterior element, with the longest facet for the radius. The three distal carpals are of approximately the same size. It is unclear how many elements distal carpal 3 and 4 articulated with distally. Distal carpal 2 articulated with the unknown anterior element proximally, and at least one metacarpal distally.

Distal to this, the fin preserves respectively seven (on the right) and five (on the left) elements, representing metacarpals and phalanges. The elements are circular or oval in dorsal view, not squared as in Platypterygius australis (Zammit, Norris & Kear, 2010), Platypterygius hercynicus (Kolb & Sander, 2009), and Sveltonectes insolitus (Fischer et al., 2011). An interesting feature is that the forefin elements seem to decrease abruptly in size distally, resulting in small metacarpals compared to the distal carpals and even smaller phalanges distal to the metacarpals. Both forefins were found in articulation, but might be missing distal elements. However, elements of such a small size occurring in the fourth and fifth row distal to the humerus is uncommon, and differ from many species with preserved forefins such as Cryopterygius kristiansenae, Platypterygius hercynicus, Aegirosaurus leptospondylus, Grendelius alekseevi, and Brachypterygius extremus (Kirton, 1983; Bardet & Fernández, 2000; Kolb & Sander, 2009; Druckenmiller et al., 2012; Zverkov, Arkhangelsky & Stenshin, 2015). Some forefins of O. icenicus and Baptanodon natans (Gilmore, 1906; Moon & Kirton, 2016) have a relatively abrupt decrease in zeugopodial and autopodial size, but the distalmost elements preserved in PMO 222.669 are even smaller than their counterparts in those species.

Two small elements were found associated with the left forefin, out of which one has the shape of half an oval and could represent the pisiform. The other is circular in dorsal view and probably a phalanx.

Vertebral column (Fig. 13)

The atlas and axis are fully fused. There is no visible suture, but if the dia- and parapophyses of the axis are assumed to be situated on the anterior margin, the two centra are of approximately the same anteroposterior length, in addition to having the same maximum mediolateral width. The anterior (Fig. 13A) and posterior articular faces of the centrum are both pentagonal with a pronounced ventral keel, resulting in an overall cordate outline, similar to the holotype of P. hoybergeti (Druckenmiller et al., 2012) and Platypterygius hercynicus (Kolb & Sander, 2009). The element is relatively mediolaterally wider compared to height, and with a more pronounced pentagonal shape in anterior view compared to that of Mollesaurus periallus (Fernández, 1999) and Arthropterygius chrisorum (Maxwell, 2010).

In lateral view (Fig. 13B), the two rib facets of the atlas are well demarcated, of which the diapophysis is confluent with the dorsal margin. The parapophysis is situated approximately midway between the dorsal and ventral margins. The facets are confluent with the anterior margin as in Arthropterygius chrisorum (Maxwell, 2010), in contrast to O. icenicus (Moon & Kirton, 2016) where the parapophysis is interpreted to be situated posteriorly on the atlas and axis part of the element. On the axis, a circular facet is situated in the dorsal half and is interpreted to be the diapophysis as in O. icenicus (Moon & Kirton, 2016). The position of the facets are similar to those seen in Platypterygius americanus (Maxwell & Kear, 2010) but differs from Athabascasaurus bitumineus where they form a continuous ridge of bone (Druckenmiller & Maxwell, 2010). The ventral margin is approximately one cm wide in anterior and posterior view and slightly concave, possibly for articulation with an intercentrum. The groove runs along entire element and is similar to that of P. hoybergeti (Druckenmiller et al., 2012).

The remaining vertebrae from the cervical and anterior dorsal region show an increase in height posteriorly (Figs. 13D–13E). The anteroposterior length varies, but in general the more posterior vertebrae are longer than those more anterior, typical of ophthalmosaurids (Buchholtz, 2001; Massare et al., 2006; Kolb & Sander, 2009). The vertebrae are approximately circular in anterior or posterior view. The neural arch of the atlas-axis complex is fused and is smaller than the other preserved neural arches. None of the neural arches are fused to the centra.

The anterior ribs (Fig. 13E) show a variable morphology in cross-section; while some display the typical ophthalmosaurid figure eight cross-section proximally and more rounded distally (Roberts et al., 2014), others have a thickened dorsal margin resulting in a T-shape in cross-section. Some ribs have longitudinal striations in the proximal part as in the Ophthalmosauridae indet. PMO 222.670 (Delsett et al., 2017), but the striations are shallower. Many ribs show signs of having been broken and healed based on the presence of calluses.

Gastralia

The gastralia are narrow and have a circular cross section.

****

Ichthyosauria De Blainville, 1835

Neoichthyosauria Sander, 2000

Thunnosauria Motani, 1999

Ophthalmosauridae Baur, 1887

Ophthalmosauridae indet.

Referred material: PMO 222.658/PMO 230.097; skull remains, forefins, partial pectoral girdle and associated vertebrae (Figs. 14–16; Table 2; Table S2).

Figure 14 Skeletal map of PMO 222.658, referred specimen of Ophthalmosauridae indet.

Abbreviations: c, coracoid; h, humerus; in, intermedium; R, radius; sca, scapula; U, ulna; ?, undetermined elements. Scale = 100 mm.

Figure 15 Pectoral girdle and forefins of PMO 222.658, referred specimen of Ophthalmosauridae indet.

Left humerus with radius, ulna and intermedium in (A), dorsal and (B), ventral view. Left humerus in (C), anterior and (D), posterior view. Coracoids in (E), ventral view. Abbreviations: ann, anterior notch; dp, dorsal process; dpc, deltopectoral crest; gf, glenoid facet; im, incomplete margin; in, intermedium; R, radius; scaf, scapular facet; U, ulna. Scale bar= 50 mm. Photo: Lene Liebe Delsett.

Figure 16 Anterior caudal vertebrae of PMO 222.658, referred specimen of Ophthalmosauridae indet.

(A), centrum from the anteriormost portion of the assemblage with elongated facet and horizontal ridge in anterior view and in (B), lateral view. (C), centrum from the posteriormost portion of the assemblage with circular facet in anterior view and (D) lateral view. Abbreviations: a, apophysis; hr, horizontal ridge; nc, neural canal. Scale bar = 50 mm. Photo: Lene Liebe Delsett.

Locality: Island of Spitsbergen, north side of Janusfjellet, approximately 13 km north of Longyearbyen, Svalbard, Norway. UTM WGS84 33X 0518844 8696066.

Horizon and stage: Slottsmøya Member, Agardhfjellet Formation, Janusfjellet Subgroup, latest Tithonian (Late Jurassic) or possibly earliest Berriasian (Early Cretaceous). 39.1 m above the echinoderm marker bed (Delsett et al., 2016).

Preservation

A partial humerus, two partial coracoids and some unidentified pieces were found disarticulated close to the other humerus, radius, ulna, and intermedium preserved associated with three smaller forefin elements, a partial coracoid, an element which might be a partial scapula or an ischiopubis and 17 caudal vertebrae (Fig. 14). The forefin elements are very well preserved, complete and in three dimensions. The vertebral centra are three dimensional and, with the exception of one vertebra, complete. Fragments from the skull, ribs and what is probably the rest of the pectoral girdle are present but preserved in an extremely poor state that prohibits further description.

Ontogeny

The elements are relatively small compared to other SML specimens considered to be adult (Druckenmiller et al., 2012; Roberts et al., 2014), but are approximately the same size of the subadult to adult K. nui holotype (Delsett et al., 2017). The smooth and finished surface on all elements support an adult ontogenetic stage (Johnson, 1977). Other ontogenetically informative features of the skull, neural spines, and forefins are not applicable (Johnson, 1977; Kear & Zammit, 2014).

Humerus (Figs. 15A–15D)

The most complete humerus is interpreted to be a left, based on McGowan & Motani (2003). The proximal end is anteroposteriorly wider than the distal end, in contrast to PMO 222.669, PMO 224.250 and O. icenicus (Moon & Kirton, 2016). The humerus is narrower anteroposteriorly compared to proximodistal length than these taxa, with a stronger constriction mid-shaft, and is more similar to that of Sveltonectes insolitus (Fischer et al., 2011) and Acamptonectes densus (Fischer et al., 2012). In dorsal view, the anterior margin is more concave than the posterior. In the proximal part the posterior margin is remarkably dorsoventrally thin.

The dorsal process originates proximally slightly posterior to the anteroposterior midpoint. It is of the same relative height as in PMO 222.669 and PMO 224.250, but more prominent than in K. nui (Delsett et al., 2017). It extends slightly distal to the proximodistal mid-point of the element similar to PMO 224.250, which is further than in U. gorodischensis and Aegirosaurus leptospondylus (Efimov, 1999a; Bardet & Fernández, 2000). The deltopectoral crest is restricted to the anterior portion of the ventral surface and is limited in size; however, it extends to the proximodistal midpoint and thus resembles PMO 222.667 and PMO 222.669, whereas it is more prominent in PMO 224.250. The distal end has three articular facets for the preaxial accessory element, radius, and ulna. Unlike PMO 222.667, PMO 224.250, U. trautscholdi, the Russian Arthropterygius sp. specimen SGM 1502 (Arkhangelsky & Zverkov, 2014; Zverkov et al., 2015; Delsett et al., 2017) and many O. icenicus specimens (Moon & Kirton, 2016), the ulnar facet is the largest as in Baptanodon natans and some specimens of Acamptonectes densus (Gilmore, 1905; Fischer et al., 2012). It is similar to Sveltonectes insolitus, but this species lacks a third facet. The facet for the preaxial accessory element is small and shallow compared to other SML ichthyosaurs (Roberts et al., 2014; Delsett et al., 2017). The ulnar facet is posteriorly deflected as in Acamptonectes densus and K. nui in contrast to Gengasaurus nicosiai and platypterygiine ophthalmosaurids (Fischer et al., 2012; Paparella et al., 2016; Delsett et al., 2017). A small area of finished bone surface is situated posterior to the ulnar facet on the posterior margin, either representing a minute facet or an area covered by articular cartilage in life.

Epipodials and remaining forefin elements (Figs. 15A–15B)

The ulna is approximately 50% larger than the radius in length and width. In PMO 222.669 the ulna is also anteroposteriorly longer than the radius, but not to such an extent. The radius and ulna of Platypterygius australis and U. trautscholdi are in contrast of similar size (Zammit, Norris & Kear, 2010; Arkhangelsky & Zverkov, 2014) whereas the ulna is significantly larger in Paraophthalmosaurus (“Yasykovia”) (Efimov, 1999b). The ulna of PMO 222.658 has a dorsoventrally tall articular facet for the humerus and appears to have little contact with the radius. The radius has a dorsoventrally tall articular side toward the humerus and a small facet for the ulna, and the facet for the intermedium is longer. The intermedium is smaller than radius and ulna. The largest of the three additional elements is oval in dorsal view. In addition, there is a smaller element with a shape similar to the possible pisiform identified in PMO 222.669. It has one straight side that is dorsoventrally short and one that is tall and concave. The third element is small and oval and likely a phalanx.

Coracoid (Fig. 15E)

The more saddle-shaped surface is interpreted to be ventral. Compared to most other ophthalmosaurids, the element is relatively flat on both surfaces (e.g., PMO 222.669, Sveltonectes insolitus and O. icenicus and (Fischer et al., 2011; Moon & Kirton, 2016)). As the coracoids are incomplete, accurate measurements are not possible, but they were probably relatively wider compared to the very mediolaterally narrow elements of N. enthekiodon and Paraophthalmosaurus (Kirton, 1983; Arkhangelsky, 1997). The intercoracoid facet is unique among ophthalmosaurids in reaching much farther anteriorly compared to the anterior margin of the scapular facet. In medial view, the intercoracoid facet is dorsoventrally tallest anteriorly. An anterior notch is present as in other ophthalmosaurids, but the size and shape of the anterior process is unknown. The glenoid and scapular facets are well demarcated, and are thus more similar to PMO 222.669 than to PMO 222.667, J. lundi and Grendelius alekseevi where the transition between the two facets is more gradual (Roberts et al., 2014; Zverkov, Arkhangelsky & Stenshin, 2015). The glenoid facet is longer than the scapular facet, but less so than in most other ophthalmosaurids, for example, U. gorodischensis (Efimov, 1999a) where the glenoid facet is considerably longer, and it resembles PMO 222.669 in this aspect (Moon & Kirton, 2016). Both the scapular and glenoid facets are triangular in lateral view, with the apex pointing anteriorly in the former, and posteriorly in the latter. In contrast the facets are fusiform in Arthropterygius chrisorum (Maxwell, 2010). The scapular facet has a more rugose surface than the glenoid facet. The articular facet of the glenoid faces slightly ventrally and not just laterally, more than in most ophthalmosaurids, and resembles PMO 222.669 in this aspect.

Vertebrae (Fig. 16)

The vertebrae are interpreted as caudals as they bear a single ventrolaterally positioned rib facet (Buchholtz, 2001; Moon & Kirton, 2016). The centra are amphicoelous, which differs from Arthropterygius chrisorum where this is absent in caudal centra (Maxwell, 2010) and is in this regard more similar to Sveltonectes insolitus (Fischer et al., 2011) and Gengasaurus nicosiai (Paparella et al., 2016). The relative proportions of height, length, and width are similar to the anterior caudal series of O. icenicus (Buchholtz, 2001) and Sveltonectes insolitus (Fischer et al., 2011). In contrast, the caudal centra have relatively higher height: length proportions in Arthropterygius chrisorum (Maxwell, 2010), whereas the ratio is slightly lower in Gengasaurus nicosiai (Paparella et al., 2016) and Athabascasaurus bitumineus (Druckenmiller & Maxwell, 2010). The eight vertebrae with the largest absolute dorsoventral height have a dorsoventrally elongated apophysis (Fig. 16B), whereas the remaining nine vertebrae have circular facets (Fig. 16D). The largest centra are interpreted to have been the anteriormost in the series (Buchholtz, 2001; Maxwell & Caldwell, 2006; Maxwell, 2010; Moon & Kirton, 2016) and the elongated facet is probably a result of its position just posterior to fusion of the diapophysis and parapophysis. When the centra are ordered based on height, there is a significant decrease in width and a small decrease in length posteriorly, which is found in many ophthalmosaurids (Massare et al., 2006; Fischer et al., 2011; Druckenmiller et al., 2012). The maximum anteroposterior length consistently appears on the ventral side of the centrum as in Platypterygius americanus possibly for a slight dorsal curvature anterior to the tail bend (Maxwell & Kear, 2010). In most vertebrae the maximum mediolateral width occurs ventral to the midpoint as in Platypterygius americanus (Maxwell & Kear, 2010), whereas some of the more posteriorly placed centra are more circular in anterior view, similar to the anterior caudal vertebrae of K. nui (Delsett et al., 2017) and Platypterygius australis (Zammit, Norris & Kear, 2010). The elongated facets of more anterior centra are more ventrally situated than the circular facets on more posterior vertebrae. This resembles the situation interpreted for? I. somersetensis (NHMUK OR14563 in McGowan & Motani, 2003; Massare & Lomax, 2018) and Platypterygius hercynicus (Kolb & Sander, 2009). The elongated and ventral facets are placed at a more equal distance between the anterior and posterior margin, whereas the circular facets touches the anterior margin. Six of the vertebrae with ventrally placed and elongated facets possess a distinct horizontal ridge dorsal to the lateral midpoint (Fig. 16B). The neural canal has dorsoventrally tall lateral margins formed by the neural arch facets. In most vertebrae, the minimum mediolateral width is encountered midway between the anterior and posterior margins but is not wholly hourglass-shaped.

***

Referred material: PMO 224.250; partial basioccipital and a nearly complete pectoral girdle and forefins (Figs. 17–20; Table 2; Table S3).

Figure 17 Skeletal map of PMO 224.250, referred specimen of Ophthalmosauridae indet. In dorsal view (stratigraphically up).

Abbreviations: c, coracoid; cl, clavicle; h, humerus; icl, interclavicle; im, interclavicle; sca, scapula. Scale = 100 mm. Modified from Delsett et al. (2016).

Figure 18 Basioccipital of PMO 224.250, referred specimen of Ophthalmosauridae indet.

The occipital condyle is shown in posterior view with the figure eight-shaped notochordal pit highlighted. Abbreviations: eca, extracondylar area; np, notochordal pit. Scale bar = 25 mm. Photo: Lene Liebe Delsett.

Figure 19 Forefins of PMO 224.250, referred specimen of Ophthalmosauridae indet.

Right humerus in (A), dorsal view with epipodials and intermedium, (D), ventral, (E), anterior and (F), posterior view. Left humerus in (B), dorsal view with radius and ulna. Unknown left forefin element in (C), dorsal (or ventral) view. Abbreviations: dp, dorsal process; dpc, deltopectoral crest; in, intermedium; pae, preaxial accessory element; paef, facet for preaxial accesory element; R, radius; Rf, facet for radius; U, ulna; Uf, facet for ulna. Scale bar = 50 mm. Photo: Lene Liebe Delsett.

Figure 20 Pectoral girdle of PMO 224.250, referred specimen of Ophthalmosauridae indet.

Articulated interclavicle and clavicles in (A), anterior view. White area on median stem is covered with fabric. Anterior portion of the right scapula in (B), lateral and (C), anterior view. Anteroventral portion of the left scapula in (D). Coracoids in (E), ventral view. Abbreviations: acp, acromion process; amp, anteromedial process; ann, anterior notch; cf, coracoid facet; gf, glenoid facet; im, incomplete margin; incf, intercarotid facet; ms, median stem; scaf, scapular facet; tb, transverse bar. Scale bar = 50 mm. Photo: Lene Liebe Delsett.

Locality: Island of Spitsbergen, Wimanfjellet, approximately 14 km north of Longyearbyen, Svalbard, Norway. UTM WGS84 33X 0523549 8696407.

Horizon and stage: Slottsmøya Member, Agardhfjellet Formation, Janusfjellet Subgroup, late Tithonian, Late Jurassic. 19 m above the echinoderm marker bed (Delsett et al., 2016).

Preservation

The clavicle-interclavicle complex is severely fractured and incomplete and the right clavicle lacks the distal tip (Fig. 17). The scapulae preserve only the anteriormost part of the element. The two coracoids are preserved in articulation, three dimensional and nearly complete, but the left is broken along the anterior margin. The humeri are complete and preserved three dimensionally. It is not possible to assign all of the forefin elements to either the left or right sides due to disarticulation. The left humerus and elements preserved closest to it are more deformed than the others. The preaxial accessory element, radius, and ulna are articulated to the right humerus. The left ulna is articulated with the humerus and the radius is found in close proximity together with an unknown element that is possibly the left preaxial accessory element. The remaining, unassigned elements vary largely in size and probably include carpals, metacarpals, and phalanges.

Ontogeny

The elements are comparable in size or larger than the largest specimens such as Cryopterygius kristiansenae and the ophthalmosaurid PMO 222.670 from the SML (Druckenmiller et al., 2012; Roberts et al., 2014), suggestive of an adult stage. As the elements are severely fractured, the surface texture is not easily detectable, but the surfaces visible on the humerus and pectoral girdle appear to be finished bone, also indicative of an advanced ontogenetic stage (Johnson, 1977).

Basioccipital (Fig. 18)

The basioccipital has a figure eight shaped notochordal pit, with the ventral loop being smaller than the dorsal. The preserved extracondylar area appears reduced compared to O. icenicus (Moon & Kirton, 2016) and more similar to P. hoybergeti and Grendelius alekseevi (Druckenmiller et al., 2012; Zverkov, Arkhangelsky & Stenshin, 2015), but is present laterally and ventrally as a peripheral groove.

Humerus (Fig. 19)

The proximal end of the humerus is slightly anteroposteriorly narrower than the distal end, as in PMO 222.669 and many other ophthalmosaurids, for example, Aegirosaurus leptospondylus, J. lundi, and O. icenicus (Bardet & Fernández, 2000; Roberts et al., 2014; Moon & Kirton, 2016) in contrast to Gengasaurus nicosiai where the distal end is anteroposteriorly wider (Paparella et al., 2016). The dorsal process is tall relative to the total dorsoventral height of the element. Its proximal end originates slightly posterior to the middle of the proximal head and ends near the approximate proximodistal midpoint of the element. As a result, the height and proximodistal length are markedly greater than that of PMO 222.669, being more similar to Platypterygius hercynicus (Kolb & Sander, 2009) and Platypterygius americanus (Maxwell & Kear, 2010). In ventral view, the deltopectoral crest is triangular in outline and of similar height to the dorsal process. It extends distally to the mid-point of the humerus. Its large size differs from PMO 222.669 and Arthropterygius chrisorum (Maxwell, 2010), which have small deltopectoral crests. It is more similar to PMO 222.667 and O. icenicus (Moon & Kirton, 2016) although not as large as in Acamptonectes densus and Sveltonectes insolitus (Fischer et al., 2011, 2012).

The distal end bears three articular facets for the preaxial accessory element, radius, and ulna typical of most ophthalmosaurids, including other SML specimens (Roberts et al., 2014; Fernández & Campos, 2015; Moon & Kirton, 2016; Paparella et al., 2016; Delsett et al., 2017) in contrast to, for example, Cryopterygius kielanae and N. enthekiodon (Kirton, 1983; Tyborowski, 2016). The facet for the preaxial element is anteriorly deflected as in, for example, Acamptonectes densus (Fischer et al., 2012), triangular in distal view and less concave than the other two facets. As in PMO 222.669, the radial facet is the largest. The ulnar facet is posteriorly deflected as in, for example, PMO 222.669 and Acamptonectes densus, in contrast to PMO 230.097, Gengasaurus nicosiai and Arthropterygius chrisorum (Maxwell, 2010; Fischer et al., 2012; Paparella et al., 2016).

Epipodium and intermedium (Figs. 19A–19C)

The preaxial accessory element is approximately circular in dorsal view and smaller than the radius and ulna. The dorsoventral height of the humeral facet is taller than the other margins. Compared to PMO 222.669, this element in PMO 224.250 is dorsoventrally taller along its anterior margin. The radius and ulna are of approximately equal size as in U. trautscholdi (Arkhangelsky & Zverkov, 2014). The facets of the radius are less well-defined than in PMO 222.669 but the radius seems to have possessed the same number of facets, with a similar relative size and general shape. The ulna varies in dorsoventral thickness, being taller proximally than distally. It is equal in anteroposterior length and proximodistal width. Similar to PMO 222.669, it has a straight posterior margin in dorsal view. Compared to PMO 222.669, the facets are less well defined, and the margins more rounded, but they are otherwise very similar in shape and in the relative size of the facets. PMO 224.250 lacks the well-defined posterior “fifth facet” on the posterior margin of the ulna as seen in PMO 222.669. The intermedium is nearly circular in dorsal view and smaller than the epipodials. It articulates with the radius and ulna proximally, and the facet for the ulna is approximately 1.0 cm long and more well-defined than other facets. The dorsoventral height is nearly uniform, but it is somewhat taller anteriorly.

Pectoral girdle

Clavicle-interclavicle complex (Fig. 20A)

The interclavicle shows the typical T-shape of ophthalmosaurids; the transverse bar is straight and probably shorter than the medial stem, as in O. icenicus (Moon & Kirton, 2016). The medial stem of the interclavicle is the best preserved portion of the element, and is almost complete. It is significantly more slender mediolaterally than PMO 222.669 and Grendelius alekseevi (Zverkov, Arkhangelsky & Stenshin, 2015). The minimum mediolateral width is encountered just posterior to the transverse bar, and posterior to this the stem widens only slightly, as in Cryopterygius kristiansenae and some specimens of O. icenicus (e.g., OUMNH J48012, L.L. Delsett, personal observation). The visceral surface is concave, but lacks the narrow trough of J. lundi (Roberts et al., 2014).

Scapula (Figs. 20B–20D)

The anterior portion is expanded dorsally and ventrally, resembling PMO 222.669. The shape of the acromion process is also similar to PMO 222.669, with a distinct dorsolateral flange that is smaller than in Sveltonectes insolitus (Fischer et al., 2011) and proportionately larger than in PMO 222.667, Cryopterygius kristiansenae and K. nui (Druckenmiller et al., 2012; Delsett et al., 2017). The glenoid and coracoid facets are not clearly separated, unlike PMO 222.669 where there is a clear separation. The coracoid facet is dorsoventrally taller than the glenoid facet. A deep dorsoventrally-oriented groove is visible primarily along the coracoid facet; it is present in both elements, and is likely not a taphonomic artefact.

Coracoid (Fig. 20E)

The two coracoids appear to be fused along the midline as in PMO 222.669. Each coracoid is slightly anteroposteriorly longer than mediolaterally wide, as in PMO 222.669 differing from the mediolaterally narrow coracoids of Sveltonectes insolitus, Paraophthalmosaurus, and N. enthekiodon (Kirton, 1983; Efimov, 1999b; Fischer et al., 2011). In ventral view, the coracoids are roughly hexagonal, with medial and lateral sides that are parallel, and the medial being the longest. The posterior margin is angled like a wide V compared to the broadly rounded or nearly straight posterior margins of J. lundi (Roberts et al., 2014) and Undorosaurus spp. (Efimov, 1999a). Cryopterygius kristiansenae (Druckenmiller et al., 2012) and Acamptonectes densus (Fischer et al., 2012) possess an angled posterior margin, however, the angle is less acute. An anterior notch is present with approximately the same relative size as in PMO 222.669 and Sveltonectes densus (Fischer et al., 2011), but mediolaterally narrower than in Acamptonectes densus (Fischer et al., 2012). The anterior process is longer than in PMO 222.669 and the anteromedial margin is angled slightly posteriorly toward the intercoracoid facet, differing from the uniquely concave margin medial to the anterior process found in J. lundi (L.L. Delsett, personal observation). The intercoracoid facets are dorsoventrally thick near the symphysis and larger ventrally than dorsally. Similar to PMO 222.669, the glenoid and scapular facets well demarcated and have a more acute angle than in Arthropterygius chrisorum (Maxwell, 2010). The scapular facet is relatively short relative to the glenoid facet when compared to PMO 222.669, but similar to the situation in Acamptonectes densus (Fischer et al., 2012) and Platypterygius hercynicus (Kolb & Sander, 2009).

Results of the landmark analysis (Fig. 21)

PC1 explains 35.0%, PC2 22.0%, and PC3 16.5%, and the remaining PCs each explain less than 10%, of the variation. Increasing values of PC1 scores describe an increasing anteroposterior length: mediolateral width and a mediolaterally narrower posterior portion, an increasing demarcation of the glenoid and scapular facets, as well as an increasingly mediolaterally narrower anterior notch. PC2 is related to the relative size of the anteromedial process, and PC3 describes mostly the shape of the anteromedial process. In the discriminant analysis, the groups with specimens of Ophthalmosaurus and of Stenopterygius are discriminated, and the MANOVA test find the two groups to be significantly different (MANOVA: p = 0.0001).

Figure 21 Principle component analysis of baracromian coracoids, delimited by the first two axes.

Morphospace for Ophthalmosaurus and Stenopterygius shown as convex hulls. Coracoid outlines shown for four example spcecimens. Specimen number and taxon see Table 1.

PC1 manages to separate coracoids that have a distinctly different outline; namely, specimens with a more rounded, equal-sided coracoid can be distinguished from those that are longer than wide, and narrower in the posterior half than in the anterior half. The former are represented by Arthropterygius chrisorum, Acamptonectes densus, PMO 222.667, and O. icenicus specimens. The latter type is found in Sveltonectes insolitus and Paraophthalmosaurus kabanovi, and in the SML material PMO 222.658.

Principal component analysis is a quantitative approach to the comparison of coracoid outlines, but the selection of landmarks might introduce some subjectivity, even when homologous points are used, and a possible extension of the analysis is the use of sliding landmarks. Among ophthalmosaurids, many species are represented by a single specimen, which is a challenge for the investigation of intraspecific or intrageneric variation. Using this method on a larger number of the taxa with many species could be an important next step to understand the phylogenetic importance of the coracoid. One limitation with PCA is that the coracoids need a complete or almost complete outline, a requirement that excludes specimens where elements are broken, such as the holotype of K. nui (PMO 222.655) (Delsett et al., 2017), or covered by other elements. A possible source of error when both dorsal and ventral surfaces are included are those specimens where the intercoracoid facet is strongly inclined.

Discussion

Taxonomic referral of the new material

PMO 222.669 is referred to P. hoybergeti, an ophthalmosaurid previously described from the SML (Druckenmiller et al., 2012). The assignment is based on a large number of similarities, most notably the presence of a greatly enlarged parietal foramen (Fig. 7), the autapomorphy of this species. It also shares most other features, including a strongly bowed jugal (Fig. 8B), a gracile and constricted stapedial shaft (Fig. 9G), a facet for preaxial accessory element on humerus (Fig. 12) and a strikingly similar basioccipital (Figs. 9A–9D). Additionally, previously undescribed material from the holotype (Fig. 4) is similar to the new specimen. Lastly, both the holotype specimen and PMO 222.669 are nearly identical in size and were recovered within four m of one another stratigraphically (Fig. 2). While both specimens are referable to the same species, the exoccipital of PMO 222.669 differs from the holotype (SVB 1451) in two aspects. In PMO 222.669 the exoccipital has two foramina while foramina are lacking in the holotype. An elongated ventral facet on the exoccipital is present in PMO 222.669, but it was not observed in the holotype. Given that the preservation of this element is far better in PMO 222.669, these differences are ascribed to taphonomic artefacts or it could result from individual variation (Maisch, 1997) and the mention of these traits have been removed from the diagnosis. The angular of PMO 222.669 (Figs. 10G and 10H) seems to lack the long anterior extension found in most ophthalmosaurids, whereas it is present in the holotype of P. hoybergeti (SVB 1451) (Druckenmiller et al., 2012). The lateral head of the stapes is slightly relatively smaller in PMO 222.669 than in the holotype (Druckenmiller et al., 2012).

The two additional specimens described above, PMO 222.658 and PMO 224.250, are confidently referred to Ophthalmosauridae on the basis of possessing a humerus (Figs. 15 and 19) with a plate-like dorsal ridge (Moon, 2017) and lacking a tuberosity on the anterodistal extremity of the humerus as well as lacking notching on forefin elements (Moon & Kirton, 2016). PMO 222.658 represents a relatively small ophthalmosaurid that shares some humeral characters with the SML ophthalmosaurid K. nui (Delsett et al., 2017), including a relatively small deltopectoral crest, a preaxial facet for an accessory element, the absence of humerus-intermedium contact and a posteriorly deflected ulnar facet. In PMO 222.658 the proximodistal length of the humerus is similar (18 mm longer) to that of the holotype of K. nui and the two specimens were found only 5.7 m apart stratigraphically. However, the specimens also display differences: The radial facet is slightly larger than the ulnar facet in K. nui while the reverse is true in PMO 222.658, although this might vary intraspecifically (Moon & Kirton, 2016; Lomax, Massare & Mistry, 2017). The preaxial accessory element facet is relatively larger in K. nui, and the relative anteroposterior length of the distal compared to the proximal end of the humerus is different. The intercoracoid facet of the coracoid (Fig. 15E) reaches further anteriorly in PMO 222.658 than in K. nui, resulting in a markedly different outline and the anterior caudal vertebrae have a height: length relationship of 2.1–3.0, less than in K. nui (contra Delsett et al., 2017). Diagnostic features in the hindfin and pelvic girdle cannot be assessed. Given both the differences and similarities to K. nui, it is not possible to assign PMO 222.658 to this taxon, especially considering that several of these characters are shared with other ophthalmosaurid taxa (e.g., Acamptonectes densus, O. icenicus, Baptanodon natans, and Arthropterygius chrisorum) (Gilmore, 1905; Maxwell, 2010; Fischer et al., 2012; Moon & Kirton, 2016).

PMO 224.250 shares features with many ophthalmosaurids such as O. icenicus, Cryopterygius kristiansenae, and Acamptonectes densus (Druckenmiller et al., 2012; Fischer et al., 2012; Moon & Kirton, 2016) but is not sufficiently complete for a generic or species assignment. An interesting feature is its size, as it represents the remains of one of the largest ichthyosaur specimens from the SML (Druckenmiller et al., 2012; Roberts et al., 2014; Delsett et al., 2017). It is not possible to make a direct size comparison with PMO 222.670, a large Ophthalmosauridae indet. specimen having an approximate total length of six m (Delsett et al., 2017) due to a lack of overlapping material, but based on a comparison to the complete holotype specimen of Cryopterygius kristiansenae, which is 5.5 m (Druckenmiller et al., 2012), PMO 224.250 is either of a similar size to PMO 222.670 or slightly larger.

New data on the cranial morphology of P. hoybergeti

The new specimen of P. hoybergeti, PMO 222.669, is more complete than the holotype specimen and helps to clarify and expand the list of diagnostic characters of this taxon. The most conspicuous feature of P. hoybergeti is the large egg-shaped parietal (“pineal”) foramen (Fig. 7), which is far larger in relative size than that known in any ophthalmosaurid. A specimen of Eurhinosaurus longirostris (based on exhibited specimen SNSB-BSPG; L.L. Delsett, personal observation) and a cf. Leptonectes specimen also possess enlarged parietal foramina that are egg-shaped and bilobed, respectively (Vincent et al., 2014). There is no clear trend through time for the development of this feature in ichthyosaurs as parietal foramina that are relatively large but still smaller than in P. hoybergeti are found in Triassic ichthyopterygians such as Utatsusaurus (Cuthbertson, Russell & Anderson, 2014) and ichthyosaurs such as Shonisaurus sikanniensis (Nicholls & Manabe, 2004), both from British Colombia in Canada. The parietal foramen is interpreted as the opening for the parietal/pineal eye, a photoreceptive organ variably present in amniotes. In lizards, the lack of a pineal eye is significantly more likely in lower latitudes, and its presence is hypothesized to be advantageous for poikilotherms in harsh climates for synchronization of reproduction and for thermoregulation (Gundy, Ralph & Wurst, 1975). A recent study on South African Permo-Triassic therapsids did not confirm the latitudinal trend for mammal ancestors, but suggests that loss of the parietal foramen was due to increased body temperature control or nocturnality (Benoit et al., 2016). There is at present insufficient material to investigate any latitude-dependent trend or a relationship to physiology for ichthyosaurs, and from the present knowledge, this feature is restricted to a single species from the SML ichthyosaur assemblage.

Among ophthalmosaurids, complete but disarticulated skulls are rare (see exceptions in Kear, 2005; Wahl, 2009; Moon & Kirton, 2016) and information has been gained from the new referred specimen because both mandibles with a majority of the elements are intact and preserved in partial articulation. The right surangular is complete anteriorly (Fig. 10A), a feature unknown in O. icenicus (Moon & Kirton, 2016). The left and right prearticulars are also preserved and the left side preserves most of the element so that the posterior margin and dorsal extent can be observed (Figs. 10E–10F). This is significant because this element is seldom identified (Moon & Kirton, 2016). PMO 222.669 also preserves the anterior portion of the pterygoid, which is rarely preserved (Kear, 2005; Fischer et al., 2011; Moon & Kirton, 2016). The preservation of a hyoid (Fig. 8F) is also significant because this element is seldom described in detail in ichthyosaurs (Wahl, 2011; Motani et al., 2013; Moon & Kirton, 2016).

P. hoybergeti is typically recovered in the ophthalmosaurid subclade Ophthalmosaurinae, along with other SML taxa (e.g., Cryopterygius kristiansenae) and O. icenicus (Fischer et al., 2011, 2012, 2013; Delsett et al., 2017). The other ophthalmosaurid subclade, Platypterygiinae, includes near-contemporaneous taxa such as Brachypterygius and Caypullisaurus (same refs). Based on the new material, P. hoybergeti possesses the ophthalmosaurine synapomorphy of a supratemporal-stapes contact (Fischer et al., 2013). However, a basioccipital with little or no exposure of extracondylar area in posterior view without a ventral notch is typical of platypterygiinae ophthalmosaurids (Fischer et al., 2012, 2014a), and both of these traits are present in P. hoybergeti. A peripheral groove on the basioccipital is often considered an ophthalmosaurine trait and is also present in this specimen, but extremely shallow.

Taxonomic utility of the pectoral girdle

PC1 and PC2 almost separate the groups of Ophthalmosaurus and Stenopterygius specimens (Fig. 21). The morphospace is thus adjacent, but only to a small degree overlapping, and the MANOVA test proved the two groups to be statistically different. Variation within Ophthalmosaurus is relatively large, but the specimens do form a cluster. Four of the SML specimens (Figs. 21–22) fall outside this cluster, while the Ophthalmosauridae indet. PMO 224.250 and the holotype of Cryopterygius kristiansenae holotype cluster inside, together with the Acamptonectes densus and U. gorodischensis holotypes. Except from this, the SML specimens are well separated from each other. The coracoid of Cryopterygius kristiansenae cannot be separated from the variation within Ophthalmosaurus, as suggested earlier (Druckenmiller et al., 2012). The holotype specimens of U. gorodischensis and Cryopterygius kristiansenae share many similarities on the coracoids, which is consistent with other features, and Cryopterygius might be a junior synonym of Undorosaurus (Druckenmiller et al., 2012; Arkhangelsky & Zverkov, 2014; Delsett et al., 2017).

Figure 22 Pectoral girdle specimens from the Slottsmøya Member Lagerstätte.

Ventral view. Interclavicles and clavicles shown together for PMO 224.250. Scale bar = 50 mm.

A diagnostic character for the genus Stenopterygius is the lack or reduction of a posterior notch of the coracoids (Maxwell, 2012). This feature is also found in ophthalmosaurids in contrast to, for example, Ichthyosaurus (Maxwell, 2012; Massare & Lomax, 2017) and might explain why Stenopterygius is not separated from ophthalmosaurids in this analysis. The specimens of Stenopterygius in this analysis have a squarer outline than the ophthalmosaurids and a more demarcated glenoid facet at its posterior margin.

The analysis confirms that there is a significant degree of variation in the outline of the coracoids, as shown by the plot and the relatively low values of the principal components (Johnson, 1979; Maxwell & Druckenmiller, 2011; Moon & Kirton, 2016). Individual or intrageneric variation is known from taxa with many specimens such as Stenopterygius and Ichthyosaurus, for instance in the shape and size of the anterior notch (Johnson, 1979; Lomax & Massare, 2017; Lomax, Massare & Mistry, 2017). However, the PCA, discriminant analysis and the MANOVA test show that there is a phylogenetic signal in the coracoid shape, as argued by Lomax (2017) and Lomax, Evans & Carpenter (2018). The phylogenetic character presently in use to cover the phylogenetic variation in coracoids is the relationship between length and width (Fischer et al., 2016). This catches little of the actual variation, but is well-suited as a phylogenetic character because the scoring is relatively objective. Caution should be taken as in the more circular or square coracoids the difference between length and width may be slight, and in Stenopterygius this is not a consistent relationship (Johnson, 1979; Maxwell, 2012).

In the scapulae, the acromion process is an important feature, phylogenetically and functionally. A “prominent” acromion process is a synapomorphy of Baracromia (Fischer et al., 2013; Moon, 2017), but evaluating the relative sizes in different specimens for scoring into data matrices is a source of uncertainty. Its prominence actually results from two different factors that vary independently; first, the degree of dorsal expansion of the anterior portion of the scapula relative to the dorsal margin of the shaft, and secondly, the anterior extent of the dorsolateral flange. Sveltonectes insolitus, PMO 222.667, PMO 222.669, and Cryopterygius kristiansenae represent all four possible combinations of strong/slight dorsal expansion and large/small extent of the dorsolateral flange (Figs. 11 and 22) (Fischer et al., 2011; Druckenmiller et al., 2012). The acromion process also varies intraspecifically in O. icenicus and Acamptonectes densus, in the latter probably due to ontogeny (Fischer et al., 2012; Moon & Kirton, 2016).

Clavicles and interclavicles are rarely preserved and studied because they are more fragile than the scapula and coracoid (Johnson, 1979; McGowan & Motani, 2003). The SML specimens (Fig. 22) show that there are variations also in these elements that might be phylogenetically valuable, such as the relative length of the medial stem of the interclavicle compared to the transverse bar. However, this relationship might also vary individually (Lomax, Massare & Mistry, 2017). The shape of the median stem of the interclavicle is variable, both in terms of outline in ventral view and the presence or absence of a trough dorsally, but the amount of phylogenetic signal is unknown (Johnson, 1979; Roberts et al., 2014; Moon & Kirton, 2016).

Conclusion

Three ichthyosaur specimens from the important Late Jurassic—Early Cretaceous SML on Spitsbergen provide new information about cranial and pectoral features in ophthalmosaurids and represent valuable data points in the understanding of the ichthyosaur distribution during this critical time. The species P. hoybergeti is now better understood based on an almost complete partly disarticulated skull and a complete pectoral girdle with two forefins and two distinctly different Ophthalmosauridae indet. specimens add to the knowledge about variation in pectoral girdle and forefin morphology.

Individual variation in ichthyosaur skeletal elements is currently not well understood. To gain a more quantitative understanding of coracoid outline variation, a PCA of 2D landmarks from six SML specimens and 24 other baracromian specimens was conducted for the first time and provided a valuable input to the debate on the amount of phylogenetic value in the coracoid in post-Triassic ichthyosaurs. Future studies should aim to include a large number of specimens and use quantitative approaches to reveal phylogenetic and evolutionary patterns.

Supplemental Information

Supplemental Information 1 Palvennia hoybergeti (PMO 222.669) Measurements.

Click here for additional data file.

Supplemental Information 2 PMO 230.097 Ophthalmosauridae indet. Measurements.

Click here for additional data file.

Supplemental Information 3 PMO 224.250 Ophthalmosauridae indet. Measurements.

Click here for additional data file.

Supplemental Information 4 2D landmarks from thunnosaurian coracoids for PCA.

Click here for additional data file.

A thank you is offered to the Spitsbergen Mesozoic Research Group with special thanks to Ø. Hammer for statistical support, H. A. Nakrem, and the volunteers Ø. Enger, M. Høyberget, L. Kristiansen, B. Lund, S. Larsen, T. Wensås, and C.S. Bjorå for excavation of the specimens. M.-L. K. Funke and V. S. Engelschiøn are thanked for preparation. B. Cordonnier, I. H. Økland, D. Foffa, and D. Lomax are thanked for valuable input. For providing collection access and/or pictures of specimens thanks to N. Zverkov, M. S. Arkhangelsky, V. Fischer, B. Kear, R. Vanis (SMSS), J. L. Wujek, E. Maxwell (SMNS), M. Fernández (MOZ, MLP), M. Riley (CAMSM), E. Howlett and H. Ketchum (OUMNH), V. Ward (LEIUG), O. Rauhut (SNSB-BSPG), N. Clark (GLAHM), V.M. Efimov (UPM), K. Sherburn (MANCH), M. Evans (LEICT). The reviewers V. Fischer and D. Lomax are thanked for very valuable comments on the paper.

Institutional Abbreviations

CAMSM Sedgwick Museum of Earth Sciences, UK

CMN Canadian Museum of Nature, Canada

GLAHM The Hunterian Museum, University of Glasgow, Glasgow, UK

IRSNB Royal Belgian Institute of Natural Sciences, Brussels, Belgium

LEICT New Walk Museum and Art Gallery, Leicester, UK

LEIUG University of Leicester, MANCH Manchester Museum, UK

NHMUK Natural History Museum, UK

OUMNH Oxford University Museum, UK

PMO Natural History Museum, palaeontological collections, Oslo, Norway

SMNS Staatliches Museum für Naturkunde Stuttgart, Germany

SMSS Städtisches Museum Schloss Salder, Salzgitter, Germany

SNHM Staatliches Naturhistorisches Museum Braunschweig, Germany

SNSB-BSPG Bayerische Staatssammlung für Paläontologie und Geologie, Munich, Germany

UPM Paleontological Museum of Undory, Ul’yanovsk, Russia

Additional Information and Declarations

Competing Interests

Author Contributions

Field Study Permissions

Data Availability

The authors declare that they have no competing interests.

Lene Liebe Delsett conceived and designed the experiments, performed the experiments, analyzed the data, contributed reagents/materials/analysis tools, prepared figures and/or tables, authored or reviewed drafts of the paper, approved the final draft, preparation of specimens, museum collection visit.

Patrick Scott Druckenmiller contributed reagents/materials/analysis tools, authored or reviewed drafts of the paper, approved the final draft.

Aubrey Jane Roberts contributed reagents/materials/analysis tools, authored or reviewed drafts of the paper, approved the final draft, preparation of specimens, museum collection visit.

Jørn Harald Hurum contributed reagents/materials/analysis tools, authored or reviewed drafts of the paper, approved the final draft.

The following information was supplied relating to field study approvals (i.e., approving body and any reference numbers):

The following permits were given by the Governor of Svalbard for the excavations in 2009, 2010, 2011 and 2012: 2006/00528-13, RIS ID 3707; RIS ID: 4760 and 2006/00528-39.

The following information was supplied regarding data availability:

Skeletal measurements for the three described specimens are provided in Tables S1–S3. 2D landmarks for the baracromian coracoids are provided in Table S4. The three described specimens are housed in the palaeontological collections (PMO) in the Natural History Museum, University of Oslo.

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
