# Peer review of "A new specimen of Palvennia hoybergeti: implications for cranial and pectoral girdle anatomy in ophthalmosaurid ichthyosaurs"

_PeerJ, doi:10.7717/peerj.5776_

## Round 0.1 · original submission · Minor Revisions

- The reviewers note some areas where additional literature can be addressed in the text, particularly for morphological comparisons.
- Reviewer 1 suggests some modifications to the morphometric analysis. Please address these points in your revision.
- Please consider moving the measurement tables (or at least a subset of the measurements) into the main manuscript. This would be easier for the reader to reference. At the very least, please cite the supplemental tables in the text (I do not see them mentioned in the manuscript).
- If color versions of the bone figures are easily available, please consider using those for the paper because color tends to show the morphology a bit more clearly for many specimens. This is totally optional if it is not an easy task, though.

·

Basic reporting

Dear editor, dear authors,

First of all, I would like to disclose a conflict of interest in the sense that I have published recently and I am working with one of the authors, P.S. Druckenmiller. However, I assure you that it did not affect my review in any way.

The type of research and the comments I have made cannot be easily spread into the usual sections of PeerJ but rather from a coherent text. Hence, all my comments and suggestions are reproduced here.

I like this paper. It provides a very large amount of new data on an enigmatic taxon from the otherwise well sampled Late Jurassic. The analysis of coracoids is also an interesting direction to take. The descriptions are well written and quite informative (but see below). I have a couple of suggestions and points I would like to bring here in the review process. But none of them are major, nor do they require new analyses (except maybe regarding the coracoid analyses): mainly explanations, reformulations, and deletions. As a result, my recommendation is to accept the paper after a minor revision has been successfully carried on.

I list major and minor points below. Each refer to words or sentences that I have highlighted in a .pdf version of the paper to make it easier for the authors to spot.

Main points

1. While this paper provides a much-needed and thorough description, some important references are either totally missing or largely ignored, notably the works of colleagues who described ichthyosaurs that are often similar with the fauna from Svalbard. Indeed, many features are compared with a reduced array of species that are from radically distinct spatiotemporal settings: the middle Jurassic Ophthalmosaurus icenicus and the Early Cretaceous platypterygiines Sveltonectes insolitus and Platypterygius australis. While these are indeed some of the best known ophthalmosaurids, reducing the volume of comparisons with Late Jurassic ichthyosaurs is not advisable and might artificially convey a sense of ‘uniqueness’ for Palvennia, a taxon supported by a single autapomorphy (to be I do not doubt its validity but you need to compare it more thoroughly to other species). I strongly suggest to add the following papers to the comparative description, especially for the craniodental and appendicular regions: (Romer, 1968; Fernández, 1997, 2007, Arkhangelsky, 1999, 2001, Efimov, 1999a,b; Bardet & Fernández, 2000; Arkhangelsky & Zverkov, 2014; Zverkov et al., 2015; Zverkov, Arkhangelsky & Stenshin, 2015; Tyborowski, 2016).

2. The identification/orientation of the following bones are problematic/poorly explained:
-The element interpreted as the opisthotic (e.g. Fig 9F) possesses a very unusual shape, especially the process medial to the paroccipital process. Could please you better explain this assignment and add annotations to Fig 9F accordingly?
- Fig 9H mentions the “articular view of quadrate”, while the quadrate appears to be depicted in posterior view. The articular view is posteroventrally oblique, showing the condyle.

3. Could you please provide a list of variable features among Palvennia as a table, this is so useful!

4. The coracoid geometric morphometrics appears as the most problematic part of the MS, by far. The methods are not well explained: we are just given the name of functions used in PAST, but without knowing the exact mathematical functions (and the reasons behind these choices). The authors should also explain why they used 8 landmarks while semi-landmarks (analysis of the outline) might have been more appropriate: indeed, the position of landmarks 3, 4, and 5 is just guesswork in the case of posteriorly rounded coracoids, i.e. that these points might not be homologous from one specimen to another. A similar issue arises for points 1, 7, and 8 in the case of anteriorly rounded coracoids (which is probably why the very well known Platypterygius australis was excluded from the analyses). Also, the dataset solely samples baracromians, not the wider clade Thunnosauria and this should be clearly stated in methods and the title of figure 21.
Also, the groups used to infer the degree of variation in baracromian coracoids are not on the same taxonomic level: one group represents the spread of a single species, O. icenicus, while the other represent the spread of a couple of species within the genus Stenopterygius. It might be better to restrict the groups to the specific level to avoid ambiguous interpretations. Finally, polygons depicting groups in morphospaces are convex hulls, which by essence, lack concavities

5. The last subchapter of the discussion. L1340-L1360: this entire paragraph does not provide anything new and no clear link is made with the rest of the paper
L1362-L1369. This entire paragraph rests on the expected link between the mediodistal length of ossified digits and the mediodistal length of the forefin. It is well known that such a link is hardly present in neoichthyosaurs; at least with the data presently at hand which is mainly based on Lower Jurassic forms. Indeed, at least some ichthyosaurs have a vast series of very small distal elements that are nearly always lost but which considerably lengthen the fin (Massare, Lomax & Klein, 2015). Also, the fin clearly extends more distal than the tip of the longest digit and this extension is quite variable (Bardet & Fernández, 2000; Lingham-Soliar & Plodowski, 2007). Thus, it is presently impossible to unambiguously speculate on the fin length in most ophthalmosaurids, making this whole paragraph conjectural. I would suggest to delete this subchapter entirely because it makes the paper end in a poorly constrained, conjectural way.



Minor points

L40-41: this makes the paper appear as very incremental and thus, of lesser interest. I would suggest to rephrase this.

L109: I do not understand this part, could you please clarify?

L129-131: as currently written this sentence is unclear and not correct: the acromion process is anterodorsal and can be captured by a 2D analysis. I would suggest the following: “in contrast, ichthyosaurian scapulae have processes arranged in orthogonal planes, which reduces the efficiency of 2D methods to capture the fine variations of their morphology”.

L138: for transparency, I would suggest to clearly list which data comes from whom/which specimen.

L139: photographs taken “directly above” do not prevent distortion, especially on the margins of the picture. Could you please rephrase?

L158: I have a hard time understanding why species that are so well known thanks to amazing material such as P. australis and B. natans are not considered, while a series of much less complete/fragmented specimens from the Slottsmoya (sorry I cannot make the proper vowel on my keyboard) member are.

L223: could you please provide the references for each of the comparisons?

L224: the text mentions the plural “autapomorphies” while I can only isolate one: “very large pineal foramen”. Again, for transparency and clarity, I would suggest to clearly state that the taxon is based on a single autapomorphy. This of course does not undermine the impressive differential diagnosis you have provided.

L251. “Based on the orientation” of what? “Landing”, I do not understand; do you mean the individual entered the soft sediment belly up?

L276. “Symphysis” applies, broadly and theoretically, for any joint of bone with some kind of cartilage in between. In our field it nearly always implies the mandibular symphysis and I’m not sure it can be applied to ichthyosaurian premaxillae.

L301: The ‘contact’ – or lack thereof – of the frontal with the supratemporal fenestra can be an important feature among ophthalmosaurids. Could you describe it and compare it with other taxa?

L415: consider removing “bone”

L418: “has been” instead of “is”?

L421: the reference after Janusaurus lundi is italicized.

L474: the reference after Arthropterygius chrisorum is italicized

L501-505: The meaning of this feature is unclear, could you rephrase? Figure 9F is not very informative either; could you please add annotations? The paroccipital process is indicated in the ‘middle’ of the lateral portion of the bulk of the bone while it should be process extending (postero)dorsolaterally. If this region is a reduced paroccipital process, what is the large (sheet? of) bone that is medial to it?

L696-698: This is the case for all ophthalmosaurids, not just Palvennia and Cryopterygius.

L698-L699: This smooth region is termed the acellular cementum ring (Maxwell, Caldwell & Lamoureux, 2011, 2012)

L706-707: All ichthyosaurs have trapezoidal sclerotic plates; the following sentence is more informative.

L719-720: A similar feature has been described in Grendelius/Brachypterygius by (Zverkov, Arkhangelsky & Stenshin, 2015).

L758-760: This is one of the difficulties of comparing outlines; the coracoids of Sveltonectes insolitus are very strongly thickened medially for example; which does not appear to be the case for Palvennia.

L762: The reference after “P. autralis” is italicized.

L785-786: could you please explain this sentence a bit more? All neoichthyosaurian humeri have a mid-shaft constriction.

L800: The reference after “Ophthalmosaurus icenicus” is italicized.

L800-801. Are these similar to the bulb-like process found in Sveltonectes and some other ophthalmosaurids (notably the Late Jurassic ‘Macropterygius’ material pers. obs. MJML [the Etches collection])

L807-808: and Maiaspondylus lindoei and Grendelius (Druckenmiller & Maxwell, 2010; Zverkov, Arkhangelsky & Stenshin, 2015).

L812: The reference after “Gengasaurus nicosiai” is italicized.

L813: please add “in” after “unlike”. Platypterygiine ophthalmosaurids lack this deflection of the ulnar facet as well (those with an intermedium facet on the humerus have a similar but non-homologous situation though).

L837-838: this region of the holotype of Gengasaurus is very poorly preserved and the ulna might actually be incomplete.

L843: Could you please describe the posterior margin of the ulna more thoroughly, notably whether the ulna is posteriorly tapering or not; this can be an important character.

L848: The reference after “Cryopterygius kristiansenae” is italicized. Also, this condition is seen in many other ophthalmosaurids.

L955: Given the fact that most ophthalmosaurids humeri are known, could you please add more comparisons?

L980: similar comment.

L991: similar comment.

L1034: the correct affiliation is now NHMUK I think.

L1035: The reference after “Platypterygius hercynicus” is italicized.

L1073: Same comment as for the other indet specimen above: please add comparisons, especially with other Late Jurassic taxa. Otherwise, these specimens have very little value in the present version of the paper.

L1085. I would advise against abbreviating here, as there are some other generic name starting with a P.

L1035: The reference after “Sveltonectes insolitus” is italicized.

L1171-1175: the number of exoccipital foramina is a feature that is known to vary intraspecifically in ichthyosaurs (Maisch, 1997).

L1096-1097: as in?

L1182: The reference after “Keilhauia nui” is italicized.

L1182-1184 (and the rest of the paragraph): all the features listed here also apply to Ophthalmosaurus, Baptanodon, Arthropterygius, and, depending on how you quantify what is a “relatively small deltopectoral crest”, Cryopterygius spp., and Acamptonectes.

L1189: not sure if “relationship” is the right word here, could you please clarify?

L1215-1216 and the rest of the paragraph: this is actually known to vary in Ichthyosaurus (Vincent et al., 2014). It is also a likely ontogenetic feature among plesiosaurians (e.g. Fischer et al., 2018).

L1221: or “organ” instead of “eye”?

L1229: the use of “variable” leads to confusion as it might be interpreted that this is an (intraspecifically) variable feature. May I suggest something along the lines of: “it is clear that this feature is restricted to a single species of the SML ichthyosaur assemblage”.

L1237: The reference after “Opthalmosaurus icenicus” is italicized.

L1248: Ibid.

L1259: I might be biased here, but this subdivision is a valid result from a valid analysis. The fact that it represents a genuine cladogenetic event is what is tested and questioned by the accumulation of novel data. Given the high homoplasy in the evolution of Thunnosauria, we expect a series of reversals to take place, and this might or might not affect the architecture of the ophthalmosaurid tree. In all fairness, I also think that the story is more complex that a simple subdivision in two clades, but you need new phylogenetic results to back such kind of claim.

L1262-1263. There is an issue in the fraction of variance explained by the axes. As you surely know, the total of all axes = 100%. I think you mean that axes 4, 5, 6, 7, etc. each explain less than 13% of the variance. I would suggest to provide a table with each value.

L1263: “increasing values of PC1”. I do not understand; could you please clarify?

L1267: Latin names of taxa cannot be used as adjectives.

L1289-1290: this should be tested using proper clustering methods rather than a PCA, which in essence, just a visualization tool.

L1297-1298: this is unclear; do you mean that both are congeneric, i.e. that Cryopterygius is a junior synonym of Undorosaurus?

L1304: same comment as for line 1267.

L1305-1307: This might actually result from a bias your analysis: you mix specific and generic taxonomic levels when discussing the morphological disparity; see also main point 4 above.

L1312-1313: In all honesty, who ever doubted that coracoids are shaped by phyletic heritage rather than being a purely convergent element??

L1321: May I also suggest to cite the work who coined the name? Asking for a friend… (Fischer et al., 2013)

L1329: In Acamptonectes, it is probably ontogeny; such source for the intraspecific variation in important to highlight.

Table 1: Specimen S2: “sp.” is italicized.

Good luck,

Valentin Fischer



References mentioned in the review


Arkhangelsky MS. 1999. On a new ichthyosaur from the Callovian stage of the Volga region near Saratov. Paleontological Journal 33:88–90.
Arkhangelsky MS. 2001. On a new ichthyosaur of the genus Otschevia from the Volgian Stage of the Volga Region near Ulyanovsk. Paleontological Journal 35:629–634.
Arkhangelsky MS., Zverkov NG. 2014. On a new ichthyosaur of the genus Undorosaurus. Proceedings of the Zoological Institute RAS 318:187–196.
Bardet N., Fernández M. 2000. A new ichthyosaur from the Upper Jurassic lithographic limestones of Bavaria. Journal of Paleontology 74:503–511.
Druckenmiller PS., Maxwell EE. 2010. A new Lower Cretaceous (lower Albian) ichthyosaur genus from the Clearwater Formation, Alberta, Canada. Canadian Journal of Earth Sciences 47:1037–1053.
Efimov VM. 1999a. Ichthyosaurs of a new genus Yasykovia from the Upper Jurassic strata of European Russia. Paleontological Journal 33:92–100.
Efimov VM. 1999b. A new family of ichthyosaurs, the Undorosauridae fam. nov. from the Volgian stage of the European part of Russia. Paleontological Journal 33:174–181.
Fernández M. 1997. A new ichthyosaur from the Tithonian (Late Jurassic) of the Neuquén Basin (Argentina). Journal of Paleontology 71:479–484.
Fernández M. 2007. Redescription and phylogenetic position of Caypullisaurus (Ichthyosauria: Ophthalmosauridae). Journal of Paleontology 81:368–375.
Fischer V., Appleby RM., Naish D., Liston J., Riding JB., Brindley S., Godefroit P. 2013. A basal thunnosaurian from Iraq reveals disparate phylogenetic origins for Cretaceous ichthyosaurs. Biology Letters 9:1–6. DOI: 10.1098/rsbl.2013.0021.
Fischer V., Benson RBJ., Druckenmiller PS., Ketchum HF., Bardet N. 2018. The evolutionary history of polycotylid plesiosaurians. Royal Society Open Science 5. DOI: 10.1098/rsos.172177.
Lingham-Soliar T., Plodowski G. 2007. Taphonomic evidence for high-speed adapted fins in thunniform ichthyosaurs. Naturwissenschaften 94:65–70.
Maisch MW. 1997. Variationen im Verlauf der Gerhinnerven bei Ophthalmosaurus (Ichthyosauria, Jura). Neues Jahrbuch für Geologie und Paläontologie, Monatshefte 1997:425–433.
Massare JA., Lomax DR., Klein A. 2015. A large forefin of Ichthyosaurus from the U.K., and estimates of the size range of the genus. Paludicola 10:119–135.
Maxwell EE., Caldwell MW., Lamoureux DO. 2011. Tooth histology in the Cretaceous ichthyosaur Platypterygius australis , and its significance for the conservation and divergence of mineralized tooth tissues in amniotes. Journal of Morphology 272:129–135.
Maxwell EE., Caldwell MW., Lamoureux DO. 2012. Tooth histology, attachment, and replacement in the Ichthyopterygia reviewed in an evolutionary context. Paläontologische Zeitschrift 86:1–14.
Romer AS. 1968. An ichthyosaur skull from the Cretaceous of Wyoming. Contributions to Geology, Wyoming University 7:27–41.
Tyborowski D. 2016. A new ophthalmosaurid ichthyosaur species from the Late Jurassic of Owadów-Brzezinki Quarry, Poland. Acta Palaeontologica Polonica 61:1–13. DOI: 10.4202/app.00252.2016.
Vincent P., Taquet P., Fischer V., Bardet N., Falconnet J., Godefroit P. 2014. Mary Anning’s legacy to French vertebrate palaeontology. Geological Magazine 151:7–20. DOI: 10.1017/S0016756813000861.
Zverkov NG., Arkhangelsky MS., Pardo-Pérez J., Beznosov PA. 2015. On the Upper Jurassic ichthyosaur remains from the Russian North. Proceedings of the Zoological Institute RAS 319:81–97.
Zverkov NG., Arkhangelsky MS., Stenshin IM. 2015. A review of Russian Upper Jurassic ichthyosaurs with an intermedium/humeral contact – Reassessing Grendelius Mcgowan, 1976. Proceedings of the Zoological Institute RAS 319:558–588.

Experimental design

The type of research and the comments I have made cannot be easily spread into the usual sections of PeerJ but rather from a coherent text. Hence, all my comments and suggestions are reproduced in part 1 Basic reporting.

Validity of the findings

The type of research and the comments I have made cannot be easily spread into the usual sections of PeerJ but rather from a coherent text. Hence, all my comments and suggestions are reproduced in part 1 Basic reporting.

Additional comments

The type of research and the comments I have made cannot be easily spread into the usual sections of PeerJ but rather from a coherent text. Hence, all my comments and suggestions are reproduced in part 1 Basic reporting.

·

Basic reporting

I have added my comments to the authors in the 'General comments to author' section, below.

Clear and unambiguous, professional English used throughout.
- Yes.

Literature references, sufficient field
- Yes.

Professional article structure, figs, tables. Raw data shared
- Yes.

Self-contained with relevant results to hypotheses
- Yes

Experimental design

Original primary research within Aims and Scope of the journal.
- Yes.

Research question well defined, relevant & meaningful. It is stated how research fills an identified knowledge gap
- Yes. However, I have expanded upon this in the comments below.

Rigorous investigation performed to a high technical & ethical standard.
- Yes.

Methods described with sufficient detail & information to replicate
- Yes.

Validity of the findings

Impact and novelty not assessed. Negative/inconclusive results accepted. Meaningful replication encouraged where rationale & benefit to literature is clearly stated
- There study will be of interest to anybody working on ichthyosaurs. Specifically, for anybody interested in the Spitsbergen fauna and interested in the phylogenetic importance of the ichthyosaur coracoid.

Data is robust, statistically sound, & controlled
- Yes.

Conclusion are well stated, linked to original research question & limited to supporting results
- Not quite. All of the research is supported, but there is no conclusion section that really summarises the entire paper.

Additional comments

22.06.2018
Reviewer: Dean Lomax

Review PeerJ manuscript #28469

A new specimen of Palvennia hoybergeti: Implications for cranial and pectoral girdle anatomy in ophthalmosaurid ichthyosaurs

Firstly, I would like to congratulate the authors on a great paper. The descriptions of the specimens are quite excellent. Well done to you all. I do, however, have some issues with the paper, although most of these things are very minor, which are included in the marked-up pdf document, but I wanted to elaborate on a couple of these below.

1. Testing variation in the SML ophthalmosaurid coracoids
I really like what you have done with this. However, with the SML specimens you are, of course, limited to a small sample size, which can have its own issues (see below). You have also only used x6 Stenopterygius specimens, of which thousands of specimens are known, so you don’t capture the full/bigger picture. In total, you only use 30 specimens. In order to really grasp an understanding of this level of variation it’s always better to include a much greater sample size. Of course, I understand the limitations of this and I do not suggest you go and examine 100+ ophthalmosaurid coracoids and 100+ Stenopterygius coracoids, but this would be another (bigger) step to test the PCA and to explore the phylogenetic importance of the coracoid, which could also be extended to include even more taxa. Instead of doing this as part of the present study, you must include something along these lines otherwise you run the risk of being criticised for not using more specimens.

Bearing this in mind, and the reason why I wanted to bring the above to your attention, I would recommend you have a read of Massare and Lomax (2018), which discussed this sort of issue with respect to hindfins in Ichthyosaurus. We found that: fewer specimens = appears to show ‘distinct’ species characters; BUT more specimens however = show that this is a continuum of variation among specimens and that many ‘unique’ characters are simply the result of this variation. For example, in a smaller sample size, some of these features might have been considered as representative of a new species.

2. Results of landmark analysis
This and the following sections on the pectoral girdle seem out of place, or at the very least need to have an introductory couple of sentences and/or a sub-heading to explain this. It seems out of place because you go from the description of the material (importance of the material) to ‘Results of landmark…’

3. Ontogeny of PMO 222.669
You have not provided a distinct section on the ontogeny of PMO 222.669 like you have for the other described specimens.

4. Consistency
Although the MS is very well written, there are some inconsistencies, e.g. use of meter Vs metre Vs m. Or Slottsmøya Member Lagerstätte is used but sometimes SML.

5. Coracoid of PMO 222.669
You say that the coracoid is markedly longer than wide but I disagree. Your measurements are 260 vs 220, so not that much of a difference. Instead, would be better to say slightly longer than wide.

6. Conclusion.
I feel that you need to have a conclusion at the end of the manuscript, to really emphasise: a. the importance of your new specimen; b. importance of the indet specimens; c. your PCA and the findings therein; d. future studies that can be built from your study – i.e. as above, more taxa and coracoids added to the PCA may reveal even more important phylogenetic information.

References mentioned in the marked-up pdf
- Apologies that several of the following are papers authored or co-authored by me, but I feel they will assist with your study, as outlined in the marked-up Pdf.


Danise S, Twitchett RJ, Matts K. 2014. Ecological succession of a Jurassic shallow-water ichthyosaur fall. Nature Communications.

Lomax, D. R., Evans, M. and Carpenter, S. 2018. An ichthyosaur from the UK Triassic–Jurassic boundary: A second specimen of the leptonectid ichthyosaur Wahlisaurus massarae Lomax 2016. Geological Journal, 10.1002/gj.3155.

Lomax, D. R., De la Salle, P., Massare, J. A. and Gallois, R. 2018. A giant Late Triassic ichthyosaur from the UK and a reinterpretation of the Aust Cliff ‘dinosaurian’ bones. PLOS One, doi.org/10.1371/journal.pone.0194742.

Lomax, D. R. and J. A. Massare. 2015. A new species of Ichthyosaurus from the Lower Jurassic of west Dorset, England. Journal of Vertebrate Paleontology.

Lomax, D. R. and Massare, J. A. 2017. Two new species of Ichthyosaurus from the lowermost Jurassic (Hettangian) of Somerset, England. Papers in Palaeontology, 3, 1–20.

Lomax, D. R., Larkin, N. R., Boomer, S., Dey, S. and Copestake, P. 2017. The first known neonate Ichthyosaurus communis skeleton: a rediscovered specimen from the Lower Jurassic, UK. Historical Biology.

Massare, J. A. and Lomax. D. R. 2017. A taxonomic reassessment of Ichthyosaurus communis and I. intermedius and a revised diagnosis for the genus. Journal of Systematic Palaeontology, http://dx.doi.org/10.1080/14772019.2017.1291116

Massare, J. A. and Lomax, D. R. 2018. Hindfins of Ichthyosaurus: effects of large sample size on ‘distinct’ morphological characters. Geological Magazine, doi:10.1017/S0016756818000146.

McGowan, C. 1993. A new species of large, long-snouted ichthyosaur from the English lower Lias. Canadian Journal of Earth Sciences, 30: 1197–1204.

Wahl, W. R. 2009. TAPHONOMY OF A NOSE DIVE: BONE AND TOOTH DISPLACEMENT AND MINERAL ACCRETION IN AN ICHTHYOSAUR SKULL. Paludicola, 7, 107-116.

Wahl, W. R. 2011. Hyoid structure and breathing in ichthyosaurs. Journal of Vertebrate Paleontology 31: 183A. [Poster presentation]

I look forward to seeing this paper published.

Best regards,
Dean Lomax

Visiting Scientist
School of Earth and Environmental Sciences,
The University of Manchester

---

## Round 0.2 · accepted · Accept

Thank you for your attention to the comments on the previous round of the manuscript, as outlined in your detailed response letter. In my opinion, you have sufficiently addressed all concerns.

#